

# Fluvial sedimentary deposits as carbon sinks: organic carbon pools and stabilization mechanisms across a Mediterranean catchment

María Martínez-Mena[1], María Almagro[2], Noelia García-Franco[3], Joris de Vente[1], Eloisa García[1], Carolina Boix Fayos[1].

[1] Soil Erosion and Conservation Research Group, CEBAS-CSIC, Spanish Research Council, Campus de Espinardo 30100, P.O. Box 164, Murcia, Spain
[2] BC3-Basque Centre for Climate Change, Sede Building 1, 1st floor, Scientific Campus of the University of the Basque Country, 48940, Leioa, Basque Country, Spain
[3] Technische Universität München Lehrstuhl für Bodenkunde Wissenschaftszentrum Weihenstephan Emil-Ramann-Strasse 2 /I D 85354 Freising-Weihenstephan

*Correspondence to*: María Martínez-Mena (mmena@cebas.csic.es)

**Abstract.** The role of fluvial sedimentary areas as organic carbon sinks remains largely unquantified. Little is known about mechanisms of organic carbon (OC) stabilization in alluvial sediments in semiarid and sub-humid catchments where those mechanisms are quite complex because sediments are often redistributed and exposed to a range of environmental conditions

in intermittent and perennial fluvial courses within the same catchment. The main goal of this study was to evaluate the contribution of transport and depositional areas as sources or sinks of $CO_2$ at the catchment scale. We used physical and chemical organic matter fractionation techniques and basal respiration rates in samples representative of the three phases of the erosion process within the catchment: i) detachment, representing the main sediment sources from forests and agricultural upland soils, as well as fluvial lateral banks; ii) transport, representing the main channel as suspended load and

bedload; and iii) depositional areas along the channel, downstream in alluvial wedges and in the reservoir at the outlet of the catchment, representative of medium and long-term residence deposits, respectively. Our results show that most of the sediments transported and deposited downstream come from agricultural upland soils and fluvial lateral bank sources, where the physico-chemical protection of OC is much lower than that of the forest soils, which are less sensitive to erosion. The protection of OC in forest soils and medium-term depositional areas (alluvial wedges) was mainly driven by physical

stabilization mechanisms, while chemical protection of OC was observed in the fluvial lateral banks. However, in the remaining sediment sources, in sediments during transport, and after deposition in long-term deposits (the reservoir), both mechanisms are equally relevant. Mineralization of the most labile OC, intra-aggregate particulate organic matter (MPOM), was predominant during transport. Aggregate formation and OC accumulation, mainly associated with macroaggregates and occluded microaggregates within macroaggregates, were predominant in depositional areas, being more protected than the

OC from the most eroding sources (agricultural soils and fluvial lateral banks). Both temporary and permanent sediment



deposits in the catchment have an important role in erosive areas, compensating OC losses from the eroded sources and functioning as C sinks.

## 1 Introduction

Soil erosion, a complex process that causes transport and deposition of sediments with accompanying soil organic carbon (SOC) (Gregorich et al., 1998; Wang et al., 2014), affects the dynamics of the terrestrial carbon (C) cycle and have important implications for the rate of C inputs into the soil (i.e. plant net primary productivity), as well as for the accumulation and stability of organic matter in soil (Berhe and Kleber, 2013). According to some authors (Van Oost et al., 2007), lateral C fluxes are key to determine the fate of soil organic carbon (SOC) at the landscape scalebut, however current estimations of those fluxes (based on soil erosion and the associated SOC content), show high variation among studies (Quinton et al., 2010; Doetterl et al., 2012). The fate of the redistributed organic carbon (OC) depends on multiple factors: i) the nature of the soil organic matter being detached from different "sources" within a catchment (Nadeu et al., 2011; 2012; Kirkels et al., 2014); ii) its turnover rates during transport; iii) the type of erosion processes (selective or non-selective); iv) the connectivity and distance of travel between eroding sources and the streambed (Boix-Fayós et al., 2015; Wang et al., 2010); and v) the micro-environmental conditions under which the OC is stored in sedimentary settings (Van Hemelryck et al., 2011; Berhe and Kleber, 2013). All these factors, which affect the protection of OC against decomposition through physical and chemical mechanisms, remain considerably uncertain. Besides a combination of different techniques (isotopic, spectroscopic, and traditional wet chemistry) has been used (Wang et al., 2014, Kirkels et al., 2014, Liu et al., 2018) to determine  if the eroded OC is lost after erosional redistribution, a full understanding of the dynamics of and interactions between OC sources and sinks, in relation to soil erosion and redistribution, is still absent (Doetterl et al., 2016). . Determination of how the different OC pools are transported by erosion from different sources, as well as the effect of new soil structure formation by aggregation and OC stabilization after deposition, can partially contribute to answering this question (Doetterl et al., 2016). In fact, it is already known that soil aggregates physically protect OC from rapid decomposition by microorganisms (Razafimbelo et al., 2008; Six et al., 2000), and aggregate formation appears to be closely linked to soil C storage and stability (Barreto et al., 2009; Golchin et al., 1995, Salomé et al., 2010). Related to this, Wang et al. (2014) found that soil erosion and transport result in disaggregation and, consequently, SOC mineralization, while depositional and burial processes promote the formation of macroaggregates that protect OC from microbial decomposition.

Several authors have suggested that alluvial settings are important, non-quantified OC sinks (Wisser et al., 2013; Hoffmann et al., 2013; Ran et al., 2014) with mechanisms of OC stabilization that can be quite complex in semiarid and sub-humid areas.In these areas sediments are often redistributed and exposed to a wide range of environmental conditions (e.g., rewetting-drying cycles, high temperatures, and solar radiation) and subcatchments with ephemeral, intermittent, and perennial fluvial courses, sometimes within the same catchment (Boix-Fayos et al., 2015). Ephemeral and intermittent fluvial



courses represent more than 50% of the global river network, being the dominant type of water course in dry climates (Datry et al., 2014). In addition, it is expected that this percentage will increase worldwide in response to climate change and increased water extraction for human use (Shewe et al., 2014), because intermittent and ephemeral rivers will suffer more severe, sustained, and more frequent droughts (De Girolamo et al., 2017; Vadher et al., 2018). Thus, understanding the OC

stabilization mechanisms within these fluvial systems might have important implications for the stability of soil OC stocks as affected by soil erosion and in response to climate change. Moreover, the quantification of the mineralization rates and the assessment of the stabilization mechanisms of OC induced by soil erosion, together with the identification of source materials contributing to the sediment OC dynamics within a catchment, are key to determine C budgets, to feed and develop prediction models, and to handle soil conservation strategies at catchment scales (Liu et al., 2018).

From previous works, where the importance of soil disturbance in determining the way that OC associated with sediments is transported and deposited within the catchment were highlighted (Nadeu et al., 2011; Boix-Fayos et al, 2015), new research questions have arisen regarding the main stabilization mechanisms of this OC during transport and in several sedimentary deposits within the catchment. To answer these questions, here we take a step further, studying in depth how OC is mobilized across the catchment using a combination of physical and chemical organic matter fractionation techniques and

comparing the different sedimentary deposits with three main sediment sources (forest and agricultural upland soils, and fluvial lateral banks) to determine their relevance in OC-sediment transport and deposition. Furthermore, we estimate basal respiration rates of upland soils, fluvial lateral banks, sediments in transit, and the different deposits. Combining information on the OC associated with soil aggregates and mineral particles, as an indicator of OC physico-chemical protection, and basal respiration rates can shed light on whether OC is being accumulated or lost by erosion at the catchment scale.

The main objective of this work was to evaluate the potential of the transport and depositional areas as sources or sinks of atmospheric $CO_2$ in a subhumid Mediterranean catchment. The specific objectives were to: i) assess the changes in the aggregate size distribution and associated OC in the sediments mobilized by soil erosion processes; ii) identify the main stabilization mechanisms of OC in eroded soils and different sedimentary deposits; and iii) determine the potential sources of eroded OC in the sedimentary deposits through the catchment. We performed the study in a sub-catchment ( 111 km2) in

the headwaters of the Segura catchment in South East Spain.

To the best of our knowledge, this is the first study to include the assessment of microaggregates contained within macroaggregates and their associated OC, widely acknowledged as an indicator of the physical protection of SOC (Six, et al 2004, Denef et al., 2007, Six and Paustian, 2014), during the different phases of soil erosion at the catchment scale.

## 2 Methods

**2.1 Study area** The study area is located in the headwaters of the Segura catchment (Murcia, South East Spain), which drains to the Taibilla reservoir (Turrilla catchment) and is formed by three adjacent sub-catchments (Rogativa, Arroyo



Blanco, and Arroyo Tercero) covering a total area of 111 km2 (Fig. 1a). The dominant lithology of the catchment consists of marls, limestones, marly limestones, and sandstones of the Cretaceous, Oligocene, and Miocene. The dominant soils in the area are Lithosols, Regosols, and Cambisols (IUSS Working Group WRB, 2015). The catchment is representative of the environmental conditions in Mediterranean mountainous areas of medium altitude. This catchment receives 530 mm of

precipitation per year and has an average temperature of 13.5ºC. It has experienced important land use changes since the 1970s involving reforestations, including the construction of a dense network of check-dams for hydrological and sediment control. Socio-economic changes in the region resulted in the abandonment of agricultural activities and the recovery of the shrubland and forest. Nowadays, forests and shrublands represent approximately 80% of its area while agricultural land represents 20%. The hydrological and geomorphological effects of these catchment changes were studied in Boix-Fayos et

al., (2008), Quiñonero-Rubio et al., (2016), and Pérez-Cutillas et al., (2018). The main fluvial course studied is ephemeral with water only flowing a few times per year, during intense rain events. A deeper description of the study area is given in Boix-Fayos et al., (2015).

**2.2 Field experimental design and sampling**

The experimental design combines a fluvial geomorphological perspective and soil science analysis. Previous

geomorphological analyses of the channel and adjacent areas and of the dynamics of the fluvial morphology in the last 60 years (Boix-Fayos et al., 2007; Nadeu et al., 2011) were used to identify the main sources and sinks of sediments in the Turrilla basin. This previous analysis was the background to the combination of the sediment cascade (Hoffman et al., 2013; Boix-Fayos et al., 2015) and the erosion cycle as the experimental approach (Fig. 1b).

A sampling design was established (Fig. 1a, Fig. 1b, Table 1) that represented: (i) the eroding areas (source of sediments)

and detachment phase; (ii) the transport areas (main channel) and the main transport processes (suspended sediments and bedload); and (iii) the depositional areas (along the channel and downstream), representing the sedimentary phase with medium-term depositional areas (alluvial wedges) and long-term depositional areas (reservoir sediments).

In each area and erosion phase, a representative number of samples were taken to cover the spatial variability of soils and sediments within the catchment. At all points and depths, disturbed and undisturbed samples were taken. Table 1 gives

details of the number and depth of the samples. A total of 89 samples were analyzed:

1. In eroding areas (Fig. 1a, 1b), representing the detachment phase of the erosion cycle, two sources of sediments were sampled: (i) surface soils under the two main land uses of the catchment area: forest and agricultural land; and ii) fluvial lateral banks well connected to the channel (Fig. 1b, Table 1).

2. In transport areas, suspended sediments and bedload samples were collected in two fluvial reaches with different flow

regimes, representing an ephemeral stream and a permanent stream, respectively. Suspended sediment samples were taken



using a siphon sampler device designed to collect suspended sediment samples at different depths after a flooding event (Dielh 20018). Six samplers (1-L bottles) were spaced vertically (7.5 cm apart) and connected to an intake tube and an air vent. A limited number of samples, for which sufficient material was available to carry out all the analyses, were selected. The suspended sediment samples were taken over a period of 2 years. The bedload was sampled in the fluvial bars and the

channel of both ephemeral and permanent reaches. Both bare and vegetated fluvial bars were sampled in two periods of the year, representing dry and wet conditions.

3. At depositional sites, samples were taken in alluvial wedges behind check-dams that were installed in the catchment in the 1970s. These alluvial wedges had a depth of between 1 and 3 meters and many of them were covered by vegetation. They were considered sedimentary areas of medium-term residence times. At the outlet of the catchment the reservoir sediments

were sampled down to a depth of 3 meters, at several points. These sediments were considered to have long-term residence times. The samples used in the analysis, and their depths, are shown in Table 1.

### 2.3 Soil and sediment analyses

### 2.3.1 Water-stable soil aggregate size distribution

Water-stable soil aggregate-size separation was carried out using a modified wet sieving method adapted from Elliott (1986).

Briefly, a 100-g sample of air-dried soil, disaggregated by hand, was placed on top of a 2000-µm sieve and submerged for 5 min in deionized water at room temperature. The sieving was performed manually by moving the sieve up and down 3 cm, 50 times in 2 min, to achieve aggregate separation. Two sieves (250 and 63 µm) were used to obtain three aggregate classes: (i) >250 µm (macroaggregates; M), (ii) 63–250 µm (microaggregates; m), and (iii) <63 µm (silt plus clay-sized particles; min). The aggregate-size classes were oven dried (50ºC), weighed, and stored in glass jars at room temperature (21° C) (Fig.

2). From this, the mean weight diameter (MWD) was obtained as an indicator of aggregate stability.

Secondly, and in order to quantify the protected microaggregates contained within macroaggregates, the procedure described by Six et al. (2000) and Denef et al. (2004) was carried out (Fig. 2, square in green). A subsample (10 g) of the macroaggregates was immersed in deionized water on top of a 250-µm mesh screen, inside a cylinder. The macroaggregates were shaken together with 50 glass beads (4-mm diameter) until complete macroaggregates disruption was observed. Once

the macroaggregates had been broken up, microaggregates and other material <250 µm passed through the mesh screen, with the help of a continuous water flow to the sieve. The material retained on the 63-µm sieve (silt+clay; min) was wet sieved, to ensure that the isolated occluded microaggregates were water-stable (Six et al., 2000). These microaggregates obtained from macroaggregates (Mm) were oven-dried, at 50°C (24 h) in aluminum trays, and weighed. The material retained on top of the 250-µm mesh was considered the intra-aggregate particulate organic matter (MPOM), representing the most labile fraction.

It was separated and weighed after drying in an oven at 50ºC (Fig. 2).



### 2.3.2 Oxidation of mineral fractions

The free and occluded mineral fraction (< 63µm) obtained in steps 1 and 2 (Fig. 2) was oxidized by NaOCl to obtain a chemically-resistant C fraction (rOC) (Zimmermann et al., 2007) representing the passive pool. One gram of every mineral fraction was oxidized for 18 h at 25 °C, with 50 mL of 6% NaOCl adjusted to pH 8 with concentrated HCl. The oxidation

residue was centrifuged at 1000 g for 15 min, decanted, washed with deionized water, and centrifuged again. This oxidation step was repeated twice. The residue was dried at 40 °C and weighed.

### 2.3.3 Organic carbon and nitrogen analysis and pools ratios

The organic carbon (OC) and total nitrogen (N) concentrations were determined separately for each water-stable aggregate-size class and for the occluded microaggregates and occluded mineral fractions using an Elemental Analyzer (LECO

TRUSPEC CN, Michigan, USA), after carbonates removal using 2 M HCl. All the samples were analyzed in triplicate. When necessary, the OC concentration of each water-stable aggregate-size class, as well as that of the intra-aggregate POM and of the microaggregates occluded in the macroaggregates, was expressed on a sand-free aggregate basis. The OC content was also expressed on a soil basis, by multiplying the C concentration in each fraction by the weight proportion of that fraction:

OC content (g OC kg$^{-1}$soil) = (OC)fraction * (proportion of the fraction)soil

Where (OC)fraction is the OC concentration in each fraction and (proportion of the fraction)soil is the percentage of the OC that each fraction represents in the bulk soil. The same procedure was followed to analyze total N in each of the separated fractions.

The total OC and total N in the bulk soil were considered as the sum of the OC or total N in each separated water-stable

aggregate-size fraction:

      Total OC (or total N) = macroaggregate (M) + microaggregate (m) + mineral fraction (min)

Two ratios have been used through the manuscript:

 i) The protected OC ratio, as an indicator of the physico-chemical protection of OC (OC occluded in aggregates and occluded in mineral particles, compared to MPOM):

Protected OC =(OC-Mm+OC-Mmin)/(OC- Mpom)



Where OC-Mm and OC-Mmin refer to the OC associated with the occluded microaggregates and mineral fraction, respectively, and OC-MPOM refers to the OC associated with the intra-aggregate particulate organic matter fraction. The higher the ratio, the more protected the OC is (Fig. 2).

ii) The degree of microbial degradation index, as an indicator of the OC stability/decomposability, based on De Clercq et al.

(2015). According to this, the OC associated with the free microaggregate and mineral fractions (OC-m and OC-min, respectively) would represent the oldest OC, while the OC in the intra-aggregate particulate organic matter fraction (OC-MPOM) and in the occluded microaggregate and mineral fractions within macroaggregates (OC-Mm and OC-Mmin, respectively) would represent the youngest and intermediate OC. The degree of microbial degradation decreases in the following order: MPOM > occluded microaggregates > occluded mineral > free microaggregates > free mineral.

Degree of microbial degradation index $=$(OC- m+OC- min)/(OC-Mm+OC-Mmin+OC-Mpom)

Where OC-m and OC-min refer to the OC associated with the free microaggregates and mineral fraction, respectively, OC-Mm and OC-Mmin refer to the OC associated with the occluded microaggregates and occluded mineral fraction, respectively, and OC-MPOM refers to the OC associated with the intra-aggregate particulate organic matter fraction (Fig. 2). The higher the ratio, the older the OC is.

**2.4 Soil and sediment incubations**

Soils and sediments were incubated under controlled conditions (28 °C, 60% of the water holding capacity) for 32 days to estimate their potential OC mineralization rates (mg $CO_2$ kg-1 soil or sediment). We incubated three replicate samples of 30 g of the soil and sediment material from the different areas throughout the catchment that was prepared for the fractionation. Previously, the maximum water holding capacity for each sample was estimated in triplicate, following the procedure of

Howard and Howard (1993). Each sample was put in a hermetically-sealed flask (125 ml) with no further additives. The $CO_2$ released was measured periodically (every day for the first four days, every three days during the second week, and then weekly) using an infrared gas analyzer (CheckmateII, PBI Dansensor, Denmark) and the flasks were opened after each measurement to avoid the accumulation of $CO_2$. The moisture content of the samples was also checked periodically, but replacement of the evaporated water was not necessary during the experiment. We used linear interpolations between

sampling dates and then summed them across all dates to estimate the cumulative amount of $CO_2$ released (mineralized) after 32 days of incubation; basal soil or sediment respiration was expressed as mg $CO_2$–C $kg^{-1}$ soil per day.

**2.5 Statistical analysis**

Statistical tests to detect differences between the means of the sediment sources and sinks -representative of the eroding, transport, and deposition phases of the erosion process - were performed separately for each erosion phase and depth (when





applicable) using the non-parametric Kruskal-Wallis test for independent measurements. Significant differences were identified at the 0.05 probability level of significance. Spearman correlations were performed to explore the relationships between most of the studied variables within each erosion phase. All statistical analyses were carried out using SPSS 24.0 (SPSS Inc., Chicago, IL, USA).

## 3. Results

### 3.1 Water-aggregate size distribution and associated OC: M, m, and min

Eroding areas: On average, the forest soils had the highest percentage of total macroaggregates (M) and MWD values (Table 2) when compared to the agricultural soils and fluvial lateral banks. The agricultural soils and fluvial lateral banks showed the same distribution trend - a decrease in the percentage of aggregates with increasing aggregate size - while in the forest

soils a predominance of macroaggregates existed (Fig. 3a).

The forest soils also showed the highest content of OC associated with the macroaggregates (OC-M), followed by the agricultural soils. The agricultural soils showed higher OC in macroaggregates than the fluvial lateral banks, despite their lower percentage of macroaggregates (Fig. 3a, Fig. 3b). A decrease in OC with decreasing aggregate size was found in the forest and agricultural soils, while no differences in the OC associated with different aggregate sizes were observed in the

fluvial lateral banks (Fig. 3b).

Transport areas: A higher percentage of total macroaggregates and higher MWD, but a lower percentage of free mineral particles, were observed in the bedload, in comparison with the suspended sediments (Table 2). However, the suspended sediments had a higher OC-M content while the OC content in the free mineral fraction (OC-min) was similar between the two types of transport sediment (Fig. 3a, Fig. 3b). In addition, for the suspended sediments the distribution of aggregates

(min>m>M) and the OC associated with the largest aggregates (OC-M) were similar to that of the agricultural soils,

Depositional areas: In the alluvial wedges, the free mineral fraction predominated over the macroaggregates and microaggregates, regardless of depth. The opposite was observed when considering the OC content associated with these fractions, the order being OC-M > OC-m > OC-min (Fig. 3a, Fig. 3b), similar to the trend in the forest and agricultural soils. In addition, a decrease in the percentage of macroaggregates from the upper to the deepest layer of the alluvial wedges was

found. For the reservoir sediments, the percentage of macroaggregates and the OC-M content were highest in the upper layer, while the free mineral fraction represented the highest percentages among the distinct fractions in the deep sediment layers, and had the highest percentages across the catchment (Fig. 3a, Fig.3b).

### 3.2 Intra-aggregate particulate organic matter and associated OC: MPOM





Eroding areas: The MPOM fraction represented about 14% of the total in the forest soil, followed by the fluvial lateral banks and agricultural soils (lower than 5%). The associated OC content in the MPOM (OC-MPOM) oscillated between 3% and 0.61%, decreasing in the following order: forest soils > agricultural soils > fluvial lateral banks (Fig. 4a, Fig. 4b).

Transport areas: The percentage of the MPOM fraction in the sediments in transit (suspended and bedload sediments) was
about 10%. The OC-MPOM content in the suspended sediments and in the bedload sediments was similar and lower, respectively, when compared to those of the soils and fluvial lateral banks (Fig. 4b).

Depositional areas: In the upper sediment layer of the reservoir the percentage of the MPOM fraction was higher and the OC-MPOM content was lower than those of the alluvial wedges (Fig. 4b). In the upper sediment layer of the alluvial wedges the OC-MPOM content was higher, compared to the forest and agricultural soils (Fig. 4b), while the opposite happened at
the reservoir. In addition, the OC-MPOM content decreased with depth in both depositional areas, although this decrease was more pronounced in the case of the alluvial wedges sediments (Fig. 4b).

### 3.3 Occluded microaggregates and occluded mineral fraction within macroaggregates, and the associated OC: Mm and Mmin

Eroding areas. Although no clear differences were observed in the percentages of Mm and Mmin among the sediment
sources (due to high spatial variability), the contents of OC-Mm and OC-Mmin were highest in the forest soils (Fig. 4b).

Transport areas. The sediments in transit displayed a significant decrease (about 50%) in the Mm and Mmin percentages as well as in the OC associated with these fractions (OC-Mm and OC-Mmin) with respect to the forest soils. However, sediments in transit showed lower percentage of Mm but similar OC-Mm and OC-Mmin contents compared to the agricultural soils and fluvial lateral banks (Fig. 4a, Fig. 4b).

Depositional areas: The percentages of occluded microaggregates (Mm) at the reservoir, and of the occluded mineral fraction (Mmin) in the alluvial wedges, were higher than in the eroding areas (forest and agricultural soils and fluvial lateral banks) (Fig. 4a). However, both types of sedimentary deposit had slightly lower OC contents associated with these occluded fractions (OC-Mm and OC-Mmin) than the forest soils, but the values were similar to those obtained in the agricultural soils and fluvial lateral banks (Fig. 4a, 4b).

The percentage of the total Mm and Mmin decreased significantly with depth in both depositional areas, being more pronounced in the case of the reservoir, where the Mm percentage in the deep sediment layer was reduced up to 75% when compared to that in the upper sediment layer (Fig. 4a). The alluvial wedges had a higher OC-Mm content in the upper sediment layer, and higher OC-Mm and OC-Mmin contents in the deep sediment layer, compared to the reservoir (Fig. 4b).



### 3.4 Free and occluded mineral fraction resistant to NaClO oxidation: rOC-min and rOC-Mmin

Regarding the sediment source, the contents of rOC-min and rOC-Mmin were higher in the forest and agricultural soils than in the fluvial lateral banks, with values ranging from 0.55% to 0.13%. Interestingly, the rOC-min in the suspended sediments was comparable to that in the forest and agricultural soils, and was lower than that in the bedload sediments, which displayed

contents similar to those of the fluvial lateral banks (Table 3). It is noteworthy that the rOC was higher in the occluded fraction than in the free mineral fraction in the lateral fluvial bank, bedload, and deep layer of the reservoir (Table 3).

### 3.5 OC protection, microbial degradation, and basal respiration rates

The degree of physico-chemical protection of the OC - (OC-Mm + OC-Mmin)/OC-MPOM - could be divided into three groups: i) the reservoir, showing the highest ratios (more protection); ii) the soils and fluvial lateral banks, bedload

sediments, and alluvial wedges, with medium ratios; and iii) the suspended sediments, with the lowest value (less protection) (Table 4).

The microbial degradation index - (OC-m + OC-min)/(OC-Mm + OC-Mmin + OC-MPOM) - also displayed three groups: i) the fluvial lateral banks, bedload sediments, and deep layer of the reservoir, with higher indices (representing older OC); ii) the agricultural soils, suspended sediments, and deep layer of the alluvial wedges; and iii) the upper layers of the forest soils

and depositional sites, with the lowest indices (representing younger and intermediate OC) (Table 4).

The basal respiration (BR) rates ranged between 0.81 and 6.04 mg $CO_2$ $kg^{-1}$ $day^{-1}$ in the eroding sources, the lowest values occurring in the fluvial lateral banks (Table 4). In the sedimentary deposits, BR ranged from 0.70 to 13.8 mg $CO_2$ $kg^{-1}$ $day^{-1}$, the values being highest in the upper sediment layers of the alluvial wedges and lowest in the deep layer of the reservoir. In the transport areas, the suspended sediments had higher respiration rates than the bedload sediments being the second-

highest rate observed through the catchment (Table 4).

Higher C:N ratios were found in the upper sediment layers of the alluvial wedges, forest soils, and suspended sediments. A decrease in the C:N ratio with increasing depth occurred in soils and sediments of the alluvial wedges, but no changes with depth were observed at the reservoir  (Table 4).

### 3.6 Correlations between the different physical, chemical, and biological variables

Positive correlations between the labile fraction, OC-MPOM, and the total OC associated with macroaggregates (OC-M) were observed in all areas. However, correlations between OC-MPOM and the macroaggregates (M), micro within macroaggregates (Mm) percentage, and OC-Mm content were found in the eroding and depositional areas but not in



transport areas. Moreover, in the eroding and depositional areas (the reservoir), positive correlations were obtained between OC-MPOM and the oxidable occluded OC (Table 5).

The BR was highly and positively correlated with the OC-MPOM content in eroding and depositional areas but not in transport areas. On the other hand, negative correlations between BR and the protected OC (r = -0.40; p=0.01) and the degree

of microbial degradation index (r = -0.60, p=0.00) across the study areas were obtained.

## 4. Discussion

### 4.1 Dynamics of OC in the eroding areas

The differences among the sources of the sediments, in terms of aggregation and OC distribution within aggregates, determined the way in which the sediment and associated OC moved across the catchment which is in line with results given

by some authors reporting that aggregation considerably reduces the potential transport distance of eroded OC and hence potentially skews its re-distribution in watersheds towards terrestrial deposition (Hu et al., 2016).

The distribution of OC within aggregates (the OC content increased with increasing aggregate size) observed in the forest soils, together with the high percentage of occluded microaggregates within macroaggreegates (Mm), rich in OC, indicates a hierarchical order of aggregation in which macroaggregates are the nucleus for microaggregate formation (Oades, 1984).

Other authors (Sodhi et al. 2009; Wang et al., 2011) also reported a higher OC content in macroaggregates than in microaggregates in soils, which means that organic matter could be the major binding agent in such soils (Oades and Waters, 1991). Here, the agricultural soils displayed a lower aggregate stability and total OC content than the forest soils, but the same pattern of OC distribution within aggregates. Moreover, in the agricultural soils, the OC content in the free mineral fraction (OC-min) was higher than that in the occluded fractions (OC-Mm and OC-Mmin) (Fig. 4b). Altogether, this

indicates the perturbation of these agricultural soils by land use change, tillage, and water erosion which is also supported by the higher proportion of OC resistant to oxidation (Table 3) compared to forest soils, and indicates the high capacity of Mediterranean calcareous soils for OC stabilization in organo-mineral complexes, in which the OC is less susceptible to mineralization (Courtier-Murias et al., 2013; Trigalet et al., 2014; Garcia-Franco et al 2015). These stabilization mechanisms are common in Mediterranean areas and have been found in other sites, close to the study area (Garcia-Franco et al., 2014,

2015). From a geomorphological perspective, the agricultural soils showed a more rigorous selection of the detached OC produced by the perturbations cited above; as a consequence, the most stable, passive pool remained in these soils.

In the fluvial lateral banks, a lack of hierarchical order of aggregation was found, together with a low OC content in all the aggregate size classes, compared to the forest and agricultural soils, despite the fact that the three sources had similar percentages of M and Mm aggregates. This indicates a decoupling of aggregates and OC, as found for other soils and land

uses (Del Galdo et al., 2003; Denef et al., 2007), and contrasts with Elliott (1986), who reported that the distribution of OC



associated with the aggregate fractions is primarily controlled by the amount of soil present in the fraction. However, in our study, such decoupling is explained by the fact that the fluvial lateral banks were sampled on average at 80 cm depth, which is equivalent to a C-horizon with a lack of soil formation and slow OC accumulation being consistent with the significantly lower BR rates observed in this source of sediments, compared to those in the forest and agricultural soils, indicating very

low microbial activity at this depth (Table 4). At the fluvial reach scale, the lateral banks were well connected to the channel, providing sediments to the main fluvial channels.

### 4.2 Sediments and associated OC dynamics during transport

During transport, a strong selection of the texture and OC pools indicates that the most resistant OC was bound strongly to the mineral fraction (free and occluded), travelling the longest distances within the fluvial network.  Compared to the

bedload, the suspended sediments displayed a lower percentage of total macroaggregates (M) (30% and 10% for the bedload and suspended sediments, respectively), but a greater amount of OC associated with this fraction (decoupling), mainly as intra-aggregate particulate organic matter (MPOM) (Fig. 3a, Fig. 3b) suggesting mobilization of the most labile OC once the macroaggregates had been broken and resulting in less protection by physical and chemical processes. Other authors have reported how erosion enhances the release of easily mineralizable C encapsulated within aggregates in the mineralization

process (Six et al., 2004; Polyakov and Lal, 2008; Van Hemelryck et al., 2010; Wang et al., 2014; Nie et al., 2018) and how that labile OC is also more easily transported in suspension (Starr et al., 2000). On the other hand, the high labile OC (OC-MPOM) content in suspended sediments is consistent with the overall relatively higher BR rate and C:N ratio, and lower aggregation and OC protection index, compared to the eroding and depositional areas (Tables 2 and 4), indicating that mineralization might be predominant during sediment transport.

Depletion of the total OC in suspended and bedload sediments of about 66% and 80%, respectively, was observed when compared to the richest sediment source (forest soils) while, compared to the agricultural soils and fluvial lateral banks, enrichment of OC in the suspended sediments and depletion in the bedload were observed. . This, together with the positive correlation between total OC and clay (r=0.48; p<0.05) and the higher clay content and lower aggregation observed in sediments during transport, demonstrates the selectivity of erosion during detachment and transport, as other authors have

also reported (Starr et al., 2000).

 Textural comparison of the sediment sources and sinks indicates that coarse material with low aggregation and a low OC content is selectively transported as bedload, while the finest and most labile material with a high OC content continues in transport as suspended load. Part of this C might be i) mineralized before deposition, ii) mineralized once it has been deposited and before it is buried by following events (Stallard et al., 1998), or iii) transported longer distances. The similar

presence of stable OC (total occluded OC: OC-Mm + OC-Mmin), microbial degradation index (an indicator of OC stability) and OC resistant to oxidation (the most passive pool) in the transported sediments and the most erodible sources (agricultural



soils and fluvial lateral banks; see Table 4) indicates that the sediment came mainly from these eroding areas and that the more stable and resistant OC was not mineralized during transport. These results highlight the important role of both physical and chemical mechanisms in the protection of OC in transported sediments in semiarid and sub-humid climates, where erosion can be a significant contributor to the regulation of catchment C budgets. ,

## 4.3 Sediments and associated OC dynamics in depositional areas

Compared to the sediments (agricultural soils and fluvial lateral banks) that were the poorest sources of OC and to the transported sediments, a significant increase in the total OC, mainly associated with macroaggregates (OC-M; > 250 µm) and microaggregates within macroaggregates (OC-Mm), was observed in depositional areas. This increase was greater in the alluvial wedges than at the reservoir and supports the idea of new macroaggregate formation following the breakdown during transport. In addition, in the alluvial wedges the OC concentration in these newly formed macroaggregates was higher than that in the microaggregates within macroaggregates in the agricultural soil and fluvial lateral bank sources, suggesting that soil forming and OC sequestration processes are occurring in these depositional areas. In fact, in these depositional areas, the distribution of OC within aggregates was similar to that observed in the forest soils (the sediment sources richest in OC) showing a decrease in the OC content with decreasing aggregate size, reinforcing the idea that OC accumulation occurs in the sediments of depositional sites. The formation of new aggregates, providing physical stabilization for eroded soil organic matter in depositional positions, has been reported elsewhere (Berhe et al., 2012).

The similar (at the reservoir) or even higher (in the alluvial wedges) BR rates and enrichment of OC, compared to the agricultural soils and fluvial lateral banks, indicate that the formation of new aggregates while, the positive correlations of the active pool (MPOM) with the percentages of M and Mm at the reservoir, and with total OC-M and OC-Mm in both types of deposit, stress the role of the labile material in the activation of aggregate formation, as was observed also in the eroding areas. Thw formation of new aggregates might be favored by the microorganisms stimulated by the established vegetation of terrestrial and aquatic origin at these depositional areas (Boix-Fayos et al., 2015). Microbial induced processes of the microaggregates within larger aggregates protects the OC associated with these microaggregates by increasing its turnover time, and leads to long-term C sequestration (García-Franco et al., 2015). Strong positive correlations between the OC-MPOM and the occluded oxidable mineral fraction (rOC-Mmin minus OC-Mmin) at the reservoir, similar to the correlations in the eroding areas (Table 5), suggest that, at this site, fresh OC inputs are more rapidly transformed by microorganisms in an oxidable pool, promoting aggregate formation and supporting the idea that the absorption of OC into the mineral surfaces (silt and clay) by chemical processes could enhance physical protection and thus long-term OC preservation (Kennedy et al. 2002, Kirkles et al 2014, García Franco et al., 2015). Von Lützow et al. (2007) found that organic molecules stabilized by strong molecular interactions with mineral surfaces decomposed more slowly than OC stabilized by physical mechanisms (e.g. occlusion in soil aggregates). A lack of difference between the OC-Mm and OC-Mmin contents (Fig. 4b) indicates that





both physical and chemical processes might be important with regard to the reservoir sediments acting as a long-term residence deposit.

The higher percentages of MPOM and Mm, but lower OC content associated with them, at the reservoir, compared to the alluvial wedges (Fig. 4a, Fig. 4b), suggest differences in the temporal dynamics of aggregate formation between these two

types of sediment deposit. The aggregate formation is slower in the alluvial wedges than at the reservoir, where the OC is also protected much more from decomposition by microorganisms than in the alluvial wedges (higher values of the protected OC ratio; Table 4). In fact, at the reservoir the OC occluded in aggregates (Mm) represented about 20% more of the total OC, while BR rates were 66% lower (Table 4), than in the alluvial wedges. Altogether, these results indicate greater physico-chemical protection of the OC in the long-term residence deposits, compared to the medium-term ones.

In the deep layers of the depositional areas, the relatively lower OC, microbial activity, and MPOM were less favorable for aggregate formation and the process unfolded very slowly compared to the upper layers, which confirms the results reported by other authors (Xie et al., 2017). The reduction in BR rates with depth suggests that the OC has less chance of being released to reinforce the C sink potential of these deposits at the deep layerswhere higher concentration of OC resistant to oxidation in the occluded fraction, relative to the free mineral fraction, and a high degree of microbial degradation index

indicate higher OC stabilization (lower OC decomposability) compared to the upper layers (Table 3).

Moreover, the similarity of the microbial degradation indices for the deep sediments and the sediment sources (agricultural soils and fluvial lateral banks) and for the suspended and bedload sediments indicates once again that the sediments come from the identified sources and that the buried topsoil might not experience any substantial change before burial.

In summary, the medium and long-term depositional areas identified seem to be not only the main sinks for OC coming from

different sources, but also areas of aggregate formation due to the microorganism activity stimulated by fresh OC inputs of different origins. Our results are in agreement with those of Doetterl et al. (2016), who indicated that depositional landform positions are not only able to store large OC stocks but also preserve OC more effectively when compared to eroding landscape positions.

**Conclusions**

Physico-chemical mechanisms favoring OC stabilization in the sources or eroding areas and the connectivity between the latter and channel areas determine both the redistribution of OC within catchments and the C-dynamics. Good physico-chemical protection of OC in the original sediment sources results in better protection of OC in sediments during fluvial transport and deposition downstream. In ephemeral or intermittent Mediterranean streams similar to our study site, sediments often originate from agricultural soils and fluvial lateral banks, in which the physico-chemical protection of OC is much

lower than that of forest soils less sensitive to erosion.




Different stabilization mechanisms were detected in the different sediment sources, transported sediments, and sedimentary deposits: (i) the predominance of physical mechanisms of OC stabilization in forest soils and alluvial wedges; and (ii) the predominance of chemical protection of OC in fluvial lateral banks. In the other positions, both processes are equally important in the stabilization of OC.

The OC stored in depositional areas is even more protected and stabilized than the OC of the most active sediment sources (agricultural soils and fluvial lateral banks). New processes of soil formation in these deposits strengthen the role of these areas as C sinks. These results imply that both temporary and permanent sediment deposits within the catchment have an important role in erosive areas, compensating OC losses from the eroded sources and functioning as C sinks.

These results underline the importance of studying soil erosion, soil formation, and geomorphological processes together in
semiarid and sub-humid catchments, where intermittent fluvial courses are predominant. Good management of these environments will be a powerful tool for climate change mitigation, given the high potential of alluvial settings as C sinks.

**Data availability**

All relevant data are presented within the manuscript.

**Authors contribution**

Joris de Vente and Carolina Boix Fayos performed the field experimental design. Maria Almagro carried out the field sampling and analyse data related to the incubations; Noelia García Franco developed the organic matter fractionation techniques and Eloisa Garcia performed them at the laboratory; María Martínez-Mena prepared the manuscript with the
contributions of all the co-authors.

**Competing interests**

The authors declare that they have no conflicts of interest.

**Acknowledgements**

This work was financially supported by the project DISECO (CGL2014-55-405-R) from the Spanish Government, National Plan of Science. We thank the members of the Soil and Water Conservation Group (CEBAS-CSIC) who helped us in the field and laboratory work. María Almagro was supported by the Juan de la Cierva Program (Grant IJCI-2015-23500). Carolina Boix-Fayos was also supported by a project from the program "Salvador de Madariaga" (PRXI7/00045) (Ministry
of Education, Culture and Sport of Spain) and a project (20186/EE/17) of the Fundación Séneca (Regional Agency of Science of the Murcia Region), in the program "Jiménez de la Espada".



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





Table 1. Characteristics and representativeness of soil and sediment samples at the different morphological positions representing the different phases of the erosion process across the catchment.

| Morphological Position | Erosion phase | Sampling sites or processes | Depth (cm) | Sample (n) |
|---|---|---|---|---|
| Eroding areas (sediment sources) | Detachment | Forest soils* | 0-10 | 9 |
| | | Agricultural soils | 0-10 | 4 |
| | | Fluvial lateral bank | 80 | 5 |
| Transport areas (main channel) | Transport | Suspended load | Not applicable | 9 (5 events) |
| | | Bedload** | 0-10 | 24 |
| Depositional areas: Along the channel and downstream | Deposition | Alluvial wedges | 0-40 | 12 |
| | | | 40-80 | 4 |
| | | | 0-40 | 8 |
| | | Reservoir | 40-300 | 14 |

*Include a variety of soils representing the main land covers in the catchment: forests, shrublands and grasslands (3, 4 and 2 samples, respectively). **include alluvial bars and channel (18 and 6 samples, respectively).



Table 2. Soil texture, MWD and total organic carbon (OC) in different morphological positions across the catchment: eroding, transport and depositional areas.

| Morphological positions | Texture (%) | | | MWD (mm) | Total OC (g kg⁻¹) |
|---|---|---|---|---|---|
| | Clay | Silt | Sand | | |
| Forest soil | 17.1±1.9[b] | 51.9±3.8[a,b,c] | 30.1±5.72[b] | 0.96±0.26[c] | 22.1±8.9[b] |
| Agricultural soil | 18.9±3.2[b] | 58.6±8.5[b,c] | 22.4±11.1[b] | 0.26±0.05[b] | 6. 8±0.9[b] |
| Fluvial lateral bank | 15.9±3.5[b] | 43.2±7.3[b] | 38.2±10.0[b] | 0.39±0.08[b] | 3.4±1.07[a] |
| **Eroding areas** | **17. 2±1.5[B]** | **50.9±3.4[A]** | **30.6±4.5[A,B]** | **0.63±0.2[B]** | **13.4±4.8[B]** |
| | | | | | |
| Suspended load | 34.1±5.55[c] | 61.7±4.8[c] | 4.0±1.47[a] | 0.22±0.04[b,a] | 7.5±0.5[b] |
| Bedload | 9.9±0.6[a] | 39.1±2.3[a] | 50.4±2.9[c] | 0.41±0.05[c,b] | 4.3±0.3[a] |
| **Transport areas** | **16.5±2.4[A]** | **45.2±2.7[A]** | **37.7±4.2[B]** | **0.39±0.04[A]** | **5.2±0.3[A]** |
| | | | | | |
| Alluvial wedge surface | 10.9±0.4[a] | 69.2±1.9[c] | 19.8±2.17[b] | 0.73±0.04[c] | 12.8±1.8[b] |
| Alluvial wedge deep | 14.9±1.9[a,b] | 76.0±3.2[c] | 8.9±1.49[a] | 0.23±0.02[b] | 7.6±0.5[a] |
| **Alluvial wedges** | **11.9±0.7[A]** | **70.9±1.7[B]** | **17.1±2.04[A]** | **0.60±0.06[B]** | **11.5±1.4[B]** |
| | | | | | |
| Reservoir surface | 14.7±0.8[b] | 60.6±0.9[b,c] | 24.6±1.2[b] | 0.76±0.45[c] | 9.26±1.16[b] |
| Reservoir deep | 18.0±0.9[b] | 61.9±0.9[c] | 20.0±1.8[b] | 0.14±0.01[a] | 5.9±0.4[a,b] |
| **Reservoir** | **16.8±0.7[B]** | **61.5±0.7[A]** | **21.7±1.3[A,B]** | **0.37±0.06[A]** | **7.1±0.6[A,B]** |

Numeral values are means ± standard errors. Columns with different lowercase letters indicate significant differences among sites or processes. Columns with different uppercase letters means significant differences between big pooled groups ($p <$ 0.05), according to Kruskal Wallis test.



Table 3. OC resistant to NaOCl oxidation in the free mineral fraction (rOC-min) and occluded mineral fraction (rOC-Mmin); and contribution of rOC-min and rOC-Mmin to the total mineral fraction in the different morphological positions across the catchment: eroding, transport and depositional areas.

| | OC resistant to NaOCl oxidation | | | |
| --- | --- | --- | --- | --- |
| | rOC-min | rOC-Mmin | rOC-min | rOC-Mmin |
| | g 100 g$^{-1}$ aggregate | | Contribution to the total mineral fraction (%) | |
| **Eroding** | | | | |
| Forest soil | 0.55±0.25$^b$ | 0.35±0.11$^b$ | 15.33±4.5 | 3.58±1.25 |
| Agricultural soil | 0.29±0.01$^b$ | 0.29±0.01$^b$ | 28.87±2.6 | 1.24±0.53 |
| Fluvial lateral bank | 0.13±0.02$^{aA}$ | 0.29±0.03$^{bB}$ | 40.04±21.22 | 12.17±8.03 |
| **Transport** | | | | |
| Suspended load | 0.21±0.1$^b$ | 0.31±0.02$^b$ | 24.24±11.47 | 1.44±1.35 |
| Bedload | 0.13±0.02$^{aA}$ | 0.22±0.02$^{bB}$ | 18.71±3.01 | 0.72±0.15 |
| **Deposition** | | | | |
| Alluvial wedge surface | 0.23±0.04$^b$ | 0.29±0.02$^b$ | 27.55±6.49 | 1.56±0.41 |
| Alluvial wedge deep | 0.17±0.04$^b$ | 0.29±0.01$^b$ | 20.76±7.33 | 1.22±.05 |
| Reservoir surface | 0.19±0.04$^b$ | 0.26±0.02$^b$ | 16.92±1.39 | 2.01±0.41 |
| Reservoir deep | 0.12±0.01$^{aA}$ | 0.16±0.001$^{aB}$ | 11.93±0.94 | 0.62±0.09 |

5  Numeral values are means ± standard errors. Columns with different lower case letters indicate significant differences among sites or processes. ($p < 0.05$) according to Kruskal Wallis test. Rows with different capital letters means significant differences between OC fractions within each site or process.





Table 4. C:N ratios, basal respiration rate, and indicators of OC protection (OC-Mm+OC-Mmin/OC-M$_{POM}$) and microbial degradation rate (OC-m+OC-min/ OC-Mm+OC-Mmin+OC-M$_{POM}$) in the different morphological positions across the catchment: eroding, transport and depositional areas.

| | C:N Ratio | Basal Respiration (mg $CO_2$ –kg$^{-1}$ soil/sed day-1) | Protected OC* | Microbial degradation** |
|---|---|---|---|---|
| **Eroding areas** | | | | |
| Forest soil | 12.1±1.2[b] | 6.04±1.74[b,c] | 0.88±0.3[b] | 1.72±0.43 [a] |
| Agricultural soil | 8.8±0.7[a,b] | 2.59±0.29[b,c] | 1.22±0.7[b] | 5.86±1.41[b,c] |
| Fluvial lateral bank | 9.5±2.8[a], | 0.81±0.18[a] | 1.09±0.4[b] | 8.13±2.88[c] |
| **Transport areas** | | | | |
| Suspended load | 10.6±1.3[b] | 6.94±0.77[c] | 0.44±0.2[a] | 2. 67±0.97[b] |
| Bedload | 8.2±0.5[a] | 3.25±0.53[b] | 0.98±0.2[b] | 8.71±0.99[c] |
| **Depositional areas** | | | | |
| Alluvial wedge surface | 13.4±1.6[b] | 13.80±1.04[c] | 0.82±0.1[b] | 0.76±0.05[a] |
| Alluvial wedge deep | 8.4±2.1[a] | 2.83±1.04[a,b] | 1.40±0.3[b] | 3.21±0.48[b] |
| Reservoir surface | 8.5±0.6[a] | 4.58±1.47[b] | 5.53±2.8[c] | 1.09±0.19[a] |
| Reservoir deep | 8.9±2.3[a] | 0.70±0.02[a] | 11.47±1.8[c] | 12.58±1.36[c] |

5   Numeral values are means ± standard errors. Columns with different lower case letters indicate significant differences among sites or processes ($p < 0.05$) according to Kruskal Wallis test. * The higher the value the more protected the OC is. ;** The higher the value the older the OC is.



Table 5. Spearman correlation coefficients between the OC associated to the intra-aggregate POM (OC-M$_{POM}$) or basal respiration (BR) and aggregates percentage and associated total OC, MWD, occluded oxidable OC, CN ratios, protected OC and degree of microbial degradation index in the different geomorphological positions within the catchment: eroding, transport and depositional areas.

OC-M$_{POM}$ (g kg$^{-1}$ soil/sed)

| | Eroding areas | Transport areas | Depositional areas | |
| --- | --- | --- | --- | --- |
| | | | Alluvial wedges | Reservoir |
| Macroaggregates (%) | .82** | .04 | .23 | .77** |
| Micro within macro (%) | **.52** | **.39** | .04 | .27 |
| OC macroaggregates (g kg$^{-1}$ soil/sed) | .85** | .57** | .96** | .71** |
| OC micro within macro (g kg$^{-1}$ soil/sed) | .72** | .22 | **.58** | .58* |
| MWD | .78** | .13 | .30 | .81** |
| Occluded Oxidable OC (g Kg$^{-1}$ soil/sed) | .76** | .28 | .09 | .61** |

Basal respiration (mg C-CO$_2$ kg$^{-1}$ soil/sed day$^{-1}$)

| | Eroding areas | Transport areas | Depositional areas | |
| --- | --- | --- | --- | --- |
| | | | Alluvial wedges | Reservoir |
| Micro within macro (%) | .27 | .11 | .07 | .61** |
| OC macroaggregates (g kg$^{-1}$ soil/sed) | .57* | .16 | .95** | .55** |
| OC micro within macro (g kg$^{-1}$ soil/sed) | .46 | .27 | .13 | .52* |
| CN | .43 | .12 | .65* | -.02 |
| Occluded Oxidable OC (g Kg$^{-1}$ soil/sed) | **.51** | .10 | .16 | **.41** |
| OC-M$_{POM}$ | .73** | .10 | .68** | .75** |
| Protected OC | -.36 | .13 | -.61* | -.38 |
| Degree of microbial degradation index | -.70** | .25 | **-.67** | **-.45** |

5        *. **. bold: significant at p<0.05; p<0.001 and p< 0.10, respectively.





**Figure captions**

Figure 1A). Location of the Turrilla catchment and main sampling areas selected for sampling of soils and sediments: 1) The Taibilla reservoir at the outlet of the catchment (as representative of long term depositional areas) where reservoir sediments were sampled; 2) Permanent stream (in blue; Turrilla subcatchment) and intermittent stream (in orange; Rogativa

subcatchment). In this area alluvial bars (as bedload), suspended load, soils and lateral fluvial banks were sampled; (3) Middle and upstream areas where forest soils and alluvial wedges (as representative of medium term depositional areas behind check-dams) were sampled.

Figure 1B). Sediment cascade of the Turrilla catchment, with representation of sampling areas according to Table 1. Please notice that some depositional areas (mainly alluvial wedges behind check-dams) were also sampled in the middle and

upstream areas. Some soils were also sampled downstream.

Figure 2. Description of the main steps and physico-chemical analysis (wet-sieving and oxidation) used to obtain the different aggregate sized fractions (square in red), the intra-aggregate organic matter, the occluded microaggregates and the occluded mineral fractions within macroaggregates (square in green), and fractions resistant to oxidation (square in blue).

Figure 3A. Water-stable aggregate size distribution (g aggregate 100 g-1 soil): > 250 μm (macroaggregates: M), 63-250 μm (free microaggregates, m) and < 63 μm (free mineral fraction: min) in different geomorphological positions within the catchment: eroding, transport and depositional areas. Numeral values are means ± standard errors. Bars with different lowercase indicate significant differences of the different soils and sediments ($p < 0.05$) according to Kruskal Wallis test.

Figure 3B. Organic carbon content (g 100g-1 aggregate): > 250 μm (macroaggregates: M), 63-250 μm (free microaggregates, m) and < 63 μm (free mineral fraction: min) in different geomorphological positions within the catchment: eroding, transport and depositional areas. Numeral values are means ± standard errors. Bars with different lowercase indicate significant differences of the different soils and sediments ($p < 0.05$) according to Kruskal Wallis test.

Figure 4A. Weight percentage (g 100 g$^{-1}$ aggregate) of the fractions occluded within macroaggregates (M): intra-aggregate particulate organic matter (MPOM), occluded microaggregates (Mm) and occluded mineral fraction (Mmin) in different geomorphological positions within the catchment: eroding, transport and depositional areas. Numeral values are means ± standard errors. Bars with different lowercase indicate significant differences of the different soils and sediments ($p < 0.05$) according to Kruskal Wallis test.

Figure 4B. Organic carbon content (g 100 g$^{-1}$ aggregate) associated to the occluded fractions within total macroaggregates (%): intra-aggregate particulate organic matter (MPOM), occluded microaggregates (Mm) and occluded mineral fraction (Mmin) in different geomorphological positions within the catchment: eroding, transport and depositional areas. Numeral values are means ± standard errors. Bars with different lowercase indicate significant differences of the different soils and sediments ($p < 0.05$) according to Kruskal Wallis test.



Figure 1

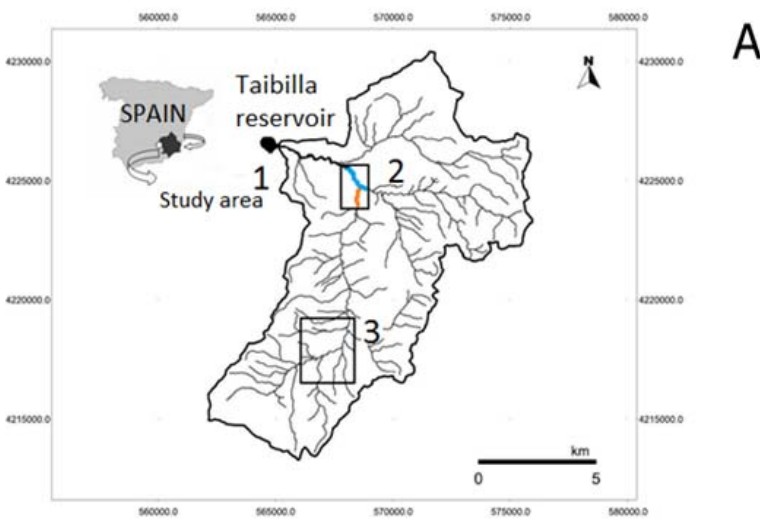

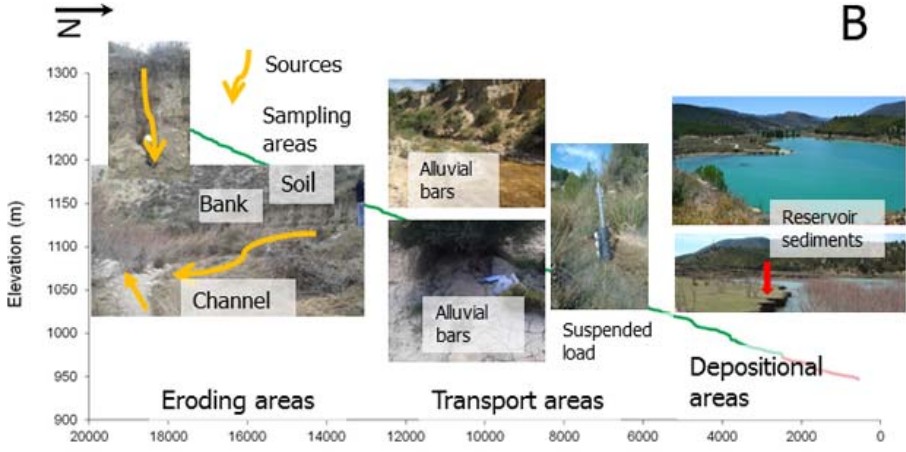



Figure 2

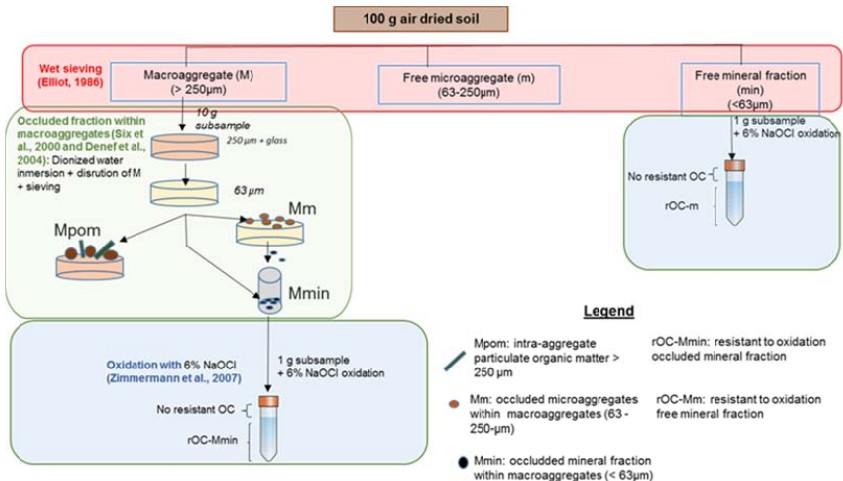





Figure 3A

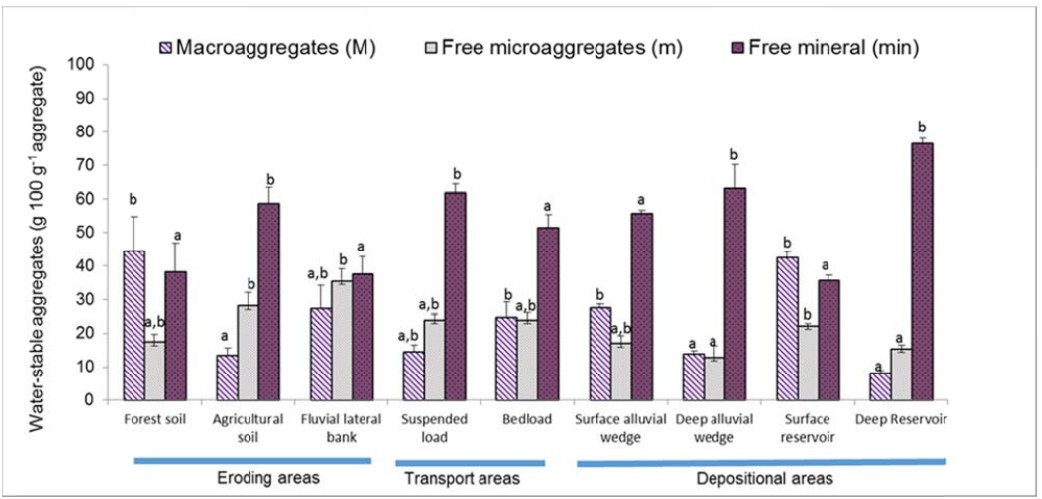





Figure 3B

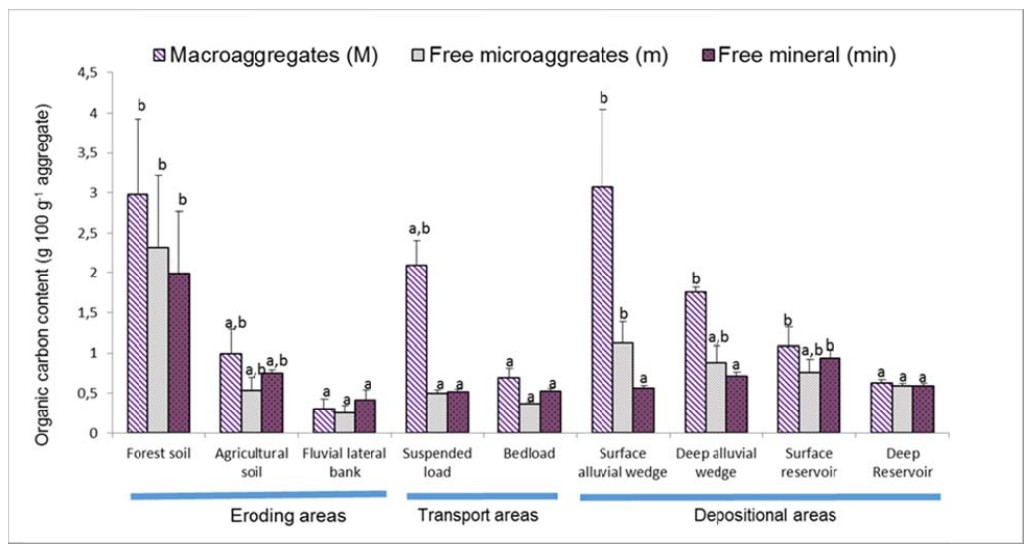

Figure 4A





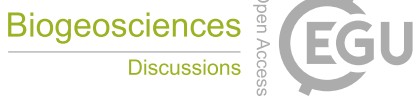

Figure 4B

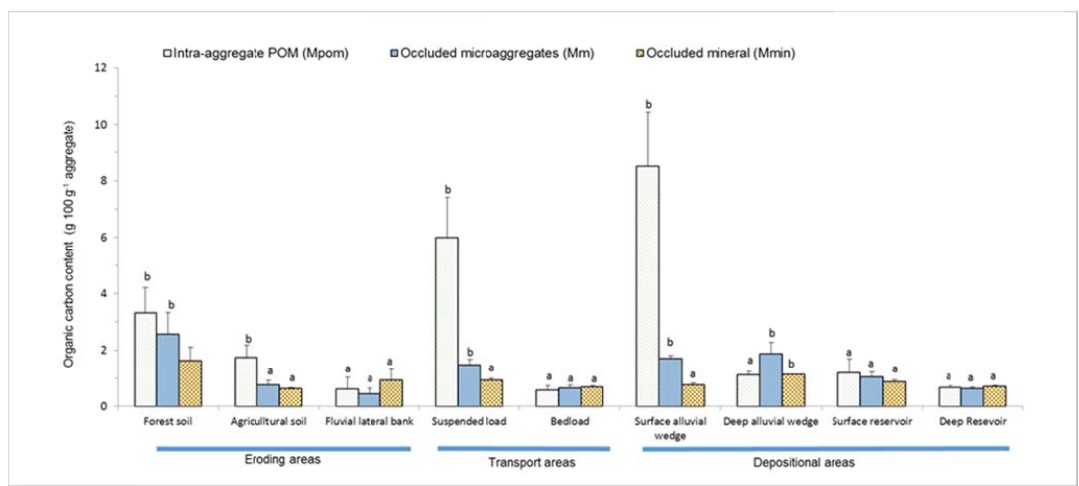