# Peer review of "Fluvial sedimentary deposits as carbon sinks: organic carbon pools and stabilization mechanisms across a Mediterranean catchment"

_Biogeosciences, 2018_

## Referee Comment (RC1) · Anonymous Referee #1 · 24 Oct 2018

bg-2018-414 presents an interesting study concerning the OC stabilization mechanisms at catchment scale by comparing the soil OC concentrations in different aggregate soil fractions in eroding, depositional and transport areas in the sub humid Mediterranean region. This paper contributes to the identification of sources and sinks of organic carbon affected by soil redistribution processes. An intense soil sample collection and a very consistent laboratory work were conducted. The main findings of the study are well supported by the data. So the results are definitely worth being published. Comments to Authors: since there are little or no binding organic compounds

with sand particles, a correction for the sand content (Elliot et al. 1991) to compare soil aggregation and OC contents should be considered. Technical corrections revise km2 and sub-humid or subhumid

---

## Author Comment (AC1) · 26 Oct 2018

Thanks very much for your comments. In relation to the sand content corrections in order to compared soil aggregation and OC contents: it was already included in the manuscript: Section 2.3.3. (page 6) when we refer: " sand-free aggregate basis". However, and in order to be more explicit we will add the reference as you suggest: Elliot et al., 1991. In relation to the technical corrections you mentioned: Km2: "2" will be write douwn as an upper suscript and sub-humid word will be unified.

---

## Editor Comment (EC1) · Park (Editor) · 26 Dec 2018

Dear Authors,

Thank you for submitting your manuscript to Biogeosciences.

To move forward the review process delayed by an assigned reviewer who does not appear to submit his report in a foreseeable time, I provide you with my own comments as detailed below. Please consider my comments in preparing your Author Responses.

[Figure]

Sincerely,

Ji-Hyung Park Associate Editor, Biogeosciences

<Major comments> 1. Terminology You cited one reference for "microbial degradation index", but I could not understand how the ratio described in the provided equation (Page 7, Line 10) may indicate the degree of microbial degradation. Furthermore, your interpretation of this ratio as an index of old C is also very confusing. As commented below, I would suggest you to articulate your rationale, supported by some data available from your or other studies. Another term "protected OC" would also require some theoretical or empirical back-ups.

2. Interpretation of results on OC in deposited sediments One of your main conclusions is stabilization of OC in deposited sediments. However, your measurements of basal respiration may indicate a higher lability of OC in your sediment samples compared with the source soils. Except some indirect indices (protected OC and microbial degradation index), you don't have any other C quality data that can support your arguments for more stable and older C in sediments. That's why you first need to provide robust backgrounds for the two ratios as well as a more in-depth discussion of the conflicting results on BR and OC compared between the source soils and sediments. In addition, your incubation settings did not consider different in situ conditions of the three set-ups (source soils, stream sediments, and deposited sediments). For example, aerobic conditions can accelerate degradation of sediment OM otherwise limited by O2? You did not provide any detail on this and other important environmental conditions such as initial moisture levels. Please discuss how the arbitrary lab conditions might deviate from the actual "filed conditions", affecting the measured rates of BR.

3 Editorial improvement Although the manuscript is generally well organized and written, I found numerous typos and scattered short paragraphs that can be more coherently organized, as some specific examples are indicated below. Please pay attention to details and revise the manuscript thoroughly.

<Specific comments> - L 24: Do you mean "representing suspended load and bedload in the main channel"? - L 30: Please define (or elaborate the meaning of) "physical stabilization" and "chemical protection". - P 2, L 1-2: Please be more specific in providing your major conclusion about the relative importance of "temporary and permanent deposits". Do you mean that both sources are equally important? - P 2, L 8: Please remove "but" and begin the following sentence with "However,". - P 3 L 10-28: Please combine these into one paragraph. - P 3 L 24: Please fix this and other "numerous" super- and subscript typos throughout the manuscript . - P 3 L 26-28: Given the importance of aggregate structure for POC stabilization during transport and deposition, you need to provide a more detailed review of the previous works on this topic. I would suggest you to expand the short introduction misplaced at the end of the first paragraph (- P 2 L 20-26) with these (Six et al„,) and more recent citations in a separate paragraph. - P 4 L 11: "more detailed" description? - P 4 L 27-28: Please provide more details on soil sampling: depth, sampling method, etc. - P 5 L 1: "after a flooding event"? As you know, the bulk of suspended load is transported "during" rainfall events, so sampling timing is a critical information. Please specify when and how long suspended sediment was collected. - P 5 L 26: What is "min"? - P 6 L6: 40oC or 50Oc? Please provide reason in case you used different temperatures. - P 6 L 25: "protected OC" is a misnomer, because this is actually a ratio of "protected OC to MPOM". - P 7 L 10: Why don't you use simply "OC-M" as denominator? In addition, it is assumed here that OC in free microaggregates and mineral fractions is older than OC in macroaggregates. Do you have any data supporting this assumption? If not, you need to reformulate relevant sentences throughout the manuscript. - P 7 L 17: 30 g soil "on a dry mass basis"? - P 7 L 14: There must have been significant reductions in soil moisture given the high incubation temperature and 32 days of incubation. Please clarify this. - P 8 L 1: Please describe why you opted for the nonparametric test. You might need to mention any prior test for normal distribution. - Sections 3.2-3.3: These two short sections can be better combined into one section, in a more coherent way to compare OC fractions among the three watershed components. - Sections 3.4 & 3.5: Please also consider
to integrate these sections in the preceding one or in a separate section on OC quality. - P 11 L 14: Please rephrase "macroaggregates are the nucleus for microaggregate formation". How can larger macroaggregates function as the nucleus? - P 11 L 8-11: I would provide an overview of major findings on different erosion sources in this beginning paragraph. - P 12 L 20-25: This type of 1 to 1 comparison between sediment and source soils does not make sense, because three sources have different source capacity. Please take into consideration estimates of source capacity in evaluating C enrichment or depletion during OC transport. - P 13 L 4: Typo at the end of the sentence. - P 13 L 6: "sediments"? - P 13 L 12: "soil forming" or "aggregate forming"? - P 13 L 17-21: Please rewrite (better split) this long and vague sentence. This and the following sentences are logically conflicting, because you are arguing that OC in the deposited sediments is more stabilized than the source soils, even though more labile materials, as evidenced by higher BR rates, exist in the same sediments. Please clarify this. - P 13 L 21: typo "Thw" - P 13 L 22: Please specify what you meant by "microbial induced processes". - P 14 L 10: You did not measure "microbial activity". - P 14 L 16-18: Again, this short paragraph can find a better place in the preceding one. - P 15 L 5: Again, you need to clarify how more stabilized OC in deposited sediments exhibited higher rates of BR compared with those measured for the source soils. - Fig 3: Please clarify in the figure legend whether significant differences indicated by different letters are among the compared fractions or soil/sediment samples.

———————————————————

---

## Author Comment (AC2) · 22 Jan 2019

Major comments

*"Terminology you cited one reference for "microbial degradation index", but I could not*
*understand how the ratio described in the provided equation (Page 7, Line 10) may indicate the*
*degree of microbial degradation. Furthermore, your interpretation of this ratio as an index of old C*
*is also very confusing. As commented below, I would suggest you to articulate your rationale,*
*supported by some data available from your or other studies".*

The microbial degradation index is based on the De Clercq et al., 2015 paper. In this study, the
authors demonstrated that the organic matter in the occluded micro-aggregate and silt clay fractions
was less degraded than the organic matter in the free micro-aggregate and silt clay fractions. They
did it by combining a fractionation method similar to ours with a stable isotope approach first
developed by Conen et al (2008).  Results from this study corroborates the aggregate formation
theory as proposed by Six et al. (2004) and Segoli et al., (2013), where the fresh residue is converted
to POM and serves as the core of newly formed macro-aggregates. Inside of these macro-aggregates,
the POM is further degraded and occluded micro-aggregates are formed. According to De Clercq et
al., 2015 interpretation, the younger and intermediate SOM is contained in the POM and occluded
fractions, while the older C is contained in the free OC fractions. Other authors (Jastrow et al., 1996;
Denef et al., 2001) have also reported that the OC associated to free particles has a slower turnover
rate than that associated to macro-aggregates, highlighting that free particles are an important
factor contributing to OC sequestration and stabilization. However, being aware of the fact that this
interpretation can be conflicting, and given that distinguishing between "old" and "young" organic
carbon was not the main aim of our approach, we have now left out this conceptual interpretation
through the manuscript. Therefore, the "microbial degradation index" is now only used as an index
that shows the high or low degree of OC stability (in terms of OC decomposability), as it has been
now mentioned in page 8 lines 6, 8, 15. Please note that many other OC quality indicators that have
been used through the manuscript (e.g., C:N ratios, basal respiration, and OC resistant to oxidation)
are consistent/support this index.

*Another term "protected OC" would also require some theoretical or empirical back-ups.*

According to literature, the OC protection mechanism can be chemical, that means "OC adhesion to
soil mineral particles", or physical, that means "particle soil aggregates that promotes the protection
of organic matter against decomposition and oxidation" (Jastrow et al., 2007; Six et al., 2002).
According to these concepts and taking into account that our results indicate a hierarchical order of
aggregation in which macroaggregates are the nucleus for microaggregate formation, and that inside
of these macro aggregates the POM is further degraded and occluded microaggregates are formed,
as it has been previously described by many authors (Six et a., 2004; Golchin et al., 1994, Segoli et al.,
2013;), we have used the "OC protection ratio" as an index to assess the different stabilization
degree of macro-aggregates present in soils and sediments at the different locations across the
catchment. This ratio is based in the "macroaggregate turnover index" by Six et al. (2000) and it is
interpreted as follows: the higher the ratio, the higher the OC is stabilized within the macro-
aggregates. Thus, we compare the two OC fractions occluded within macro-aggregates (i.e., micro-
aggregate and mineral associated OC) with the most active OC fraction (Mpom). This allows us to
have a relative estimation of how much of this active OC pool is being incorporated and stabilized
within the micro-aggregates.

Nevertheless, we have renamed this index in the revised version of the manuscript, according to the
comments made by the editor (in the text below), in order to avoid reader misunderstandings. We
have also changed: "OC protection ratio" to "macro-aggregate stabilization index" throughout the
manuscript. Moreover, we have now included the reference by Six et al (2000) in the methodology
section (page 7, lines 22-24) to clarify that the above.-mentioned index is based on this author
previous work.

*"Interpretation of results on OC in deposited sediments. One of your main conclusions is*
*stabilization of OC in deposited sediments. However, your measurements of basal respiration may*
*indicate a higher lability of OC in your sediment samples compared with the source soils. Except*
*some indirect indices (protected OC and microbial degradation index), you don't have any other C*
*quality data that can support your arguments for more stable and older C in sediments. That's why*
*you first need to provide robust backgrounds for the two ratios as well as a more in-depth*
*discussion of the conflicting results on BR and OC compared between the source soils and*
*sediments".*

Please note that our statements are not contradictory at all. On the one hand, basal respiration is a
surrogate of microbial activity and therefore can be used as an index of the lability of the material
being degraded by the microorganisms. Microorganisms are activated by the presence of labile
(easily decomposable) organic matter and then respire more. But this labile organic matter is further
transformer into other bio-products which can be more or less recalcitrant, can be chemically
attached to other molecules and even physically protected by aggregates, as it has been pointed out
by many previous studies (Six et al., 2000; 2002; Cotrufo et al 2013; Denef et al 2004, etc).At the end,
organic matter can be protected (and preserved) by their inherent chemical composition, by physical
protection mechanisms (intra-POM), or both. See Schmidt et al 2011 for further details.

Nevertheless, higher basal respiration in sediments compared to the source soils only occurred at the
surface layer of the deposits (e.g., alluvial wedges and reservoir; please see Table 4), where at the
same time aggregate formation is occurring (as explained before). On the contrary, at the deep layers
of the alluvial wedges and the reservoir, where the carbon is being stored and stabilized, much lower
basal respiration rates than at the upper layers, and even than in the source soils are observed.

*"… In addition, your incubation settings did not consider different in situ conditions of the three*
*setups (source soils, stream sediments, and deposited sediments). For example, aerobic conditions*
*can accelerate degradation of sediment OM otherwise limited by O2? You did not provide any*
*detail on this and other important environmental conditions such as initial moisture levels. Please*
*discuss how the arbitrary lab conditions might deviate from the actual "filed conditions", affecting*
*the measured rates of BR."*

We agree with the comment made by the editor that soil and sediment respiration rates under field
conditions may differ from those under laboratory controlled conditions. Indeed different techniques
(soil chambers vs laboratory incubations) are used. It´s also true that aerobic conditions can
accelerate degradation of sediment OM otherwise limited by $O_2$. However, it was necessary to run
the soil and sediment incubations under standardized controlled conditions (28 °C and at 60% of its
water holding capacity) in order to be able to make comparisons between the large variety of soils
and sediments distributed within the catchment, and exposed to different local environmental
conditions. Please note that given the large number of sampling points, sometimes located some
kilometers apart from each other, it was not possible to perform *in situ* field soil respiration
measurements with soil respiration chambers within the same period (e.g., from 10 am to 13 pm) to avoid variations in soil temperature and moisture (well-known as major environmental drivers of soil
respiration spatial and temporal variability; see Almagro et al., 2009 and references therein). It is
well-known that respiration is highly sensitive to moisture conditions, and such variations could have
biased our experiment outcomes. Moreover, this approach is not valid to estimate respiration rates
in deep soil or sediment layers. Because of all those drawbacks, we had to perform soil and sediment
incubations in the laboratory. Nevertheless, as our aim was to assess the potential microbial activity
(as a surrogate of basal respiration rates) as well as characterize the quality of the organic carbon
present in the different soils and sedimentary deposits (i.e., basal rates rates are also an indicator of
the lability or recalcitrance of OC) we had to perform the incubations under standardized controlled
conditions in order to avoid temperature and moisture variation among samples. In such a way, we
can state that the observed variability in basal respiration rates among soils and sediments are
explained by the different quality of their OC contents.

Regarding the initial moisture levels of the collected soils and sediments we estimated them before
setting the incubations and the variability among samples was huge. We used the initial value of each
sample to estimate the amount of water that had to be added to each specific soil or sediment
sample in order to achieve a 60% of its water holding capacity (likewise estimated for each soil and
sediment sample). Therefore, all samples were incubated under the same environmental conditions
to avoid bias associated with different soil moisture contents.

*3."Editorial improvement Although the manuscript is generally well organized and written, I found*
*numerous typos and scattered short paragraphs that can be more coherently organized, as some*
*specific examples are indicated below. Please pay attention to details and revise the manuscript*
*thoroughly.*

Thanks very much for your suggestion. We have now revised the manuscript changing all those
paragraphs that were not clear and corrected all typographic errors (marked on the manuscript).

*<Specific comments:*

*"L 24: Do you mean "representing suspended load and bedload in the main channel"?*

Yes, we have changed it (page 1, Line 24).

*"L 30: Please define (or elaborate the meaning of) "physical stabilization" and "chemical*
*protection".*

In our study , we have used two main mechanisms of SOC stabilization (from Six et al 2002): (1)
physical protection, which refers to the isolation of microbes and enzymes from carbon substrates by
the physical barriers of soil aggregates; and (2) chemical protection, which means the protection of
OC by binding to soil minerals.

We agree with the editor that it is necessary to clarify this terminology that can be sometimes
confusing through the manuscript because the physico-chemical protection mechanisms lead to the
organic carbon stabilization. In order to clarify the terminology and be consistent throughout the
manuscript we have now changed physical stabilization by "physical protection" and we have added
between brackets the meaning of physical and chemical protection: "OC within aggregates", and "OC
adhesion to soil mineral particles", respectively): page 1, lines 30-31.

*P 2, L 1-2: Please be more specific in providing your major conclusion about the relative importance*
*of "temporary and permanent deposits. Do you mean that both sources are equally important?"*

According to the results from our work we consider that both, temporary and permanent deposits
are very important from the point of view of the physico-chemical mechanisms of OC protection and
stabilization that are occurring across the catchment. In addition, it is not our purpose to determine
if one of them is more important than the other. We consider that both temporary and permanent
deposits should be preserved, although not beyond the natural fluvial dynamics specially for the
natural transitory deposits (short-term residence times), due to their high potential as C sinks.

*"P 2, L 8: Please remove "but" and begin the following sentence with "However,".*

It has been done

*"P 3 L 10-28: Please combine these into one paragraph". –*

It has been combined.

**"P 3 L 24: Please fix this and other "numerous" super- and subscript typos throughout the**
**manuscript"** .

Thanks and sorry for the errors. We have now revised and changed all of them. MPOM has been
unified and change by Mpom through the manuscript; Km2 has been changed to Km$^2$ (marked on the
manuscript).

**"P 3 L 26-28: Given the importance of aggregate structure for POC stabilization during transport**
**and deposition, you need to provide a more detailed review of the previous works on this topic. I**
**would suggest you to expand the short introduction misplaced at the end of the first paragraph (- P**
**2 L 20-26) with these (Six et al,,,) and more recent citations in a separate paragraph".**

Thanks very much. Your suggestion has been accepted. We have now moved the paragraph,
extended our arguments and new references have been added  to support our statements  in the
introduction: Hoffman et al., 2013 (page 2, line 20) Lal, 2005; Boix Fayos et al., 2015; Berhe et al.,
2013,  2018 ; Nie et al., 2018. See page 2 lines 26-31.

*P 4 L 11: "more detailed" description? –*

Because there are many previous works in this study area (already cited in this section (Boix-Fayos et
al., 2007, 2015,), Quiñonero-Rubio et al., (2016), and Pérez-Cutillas et al., (2018)), we consider that
extend the description could revert in repeating information already published.

*P 4 L 27-28: Please provide more details on soil sampling: depth, sampling method, etc. –*

More details on soil sampling has been included

*P 5 L 1: "after a flooding event"? As you know, the bulk of suspended load is transported "during"*
*rainfall events, so sampling timing is a critical information. Please specify when and how long*
*suspended sediment was collected. –*

We have specified that the sampling was done immediately after each flooding event (page 7, line 7).

*P 5 L 26: What is "min"? –*

Sorry for the abbreviation. It means minutes. We have changed it (page 6, lines 13-14).

**P 6 L6: 40oC or 50Oc? Please provide reason in case you used different temperatures.**

*Sorry, it was a mistake. 50ºC is the temperature for both analyses (page 7, line 7).*

*P 6 L 25: "protected OC" is a misnomer, because this is actually a ratio of "protected OC to MPOM".*

This has been already responded in "major comments" point 2.

*P 7 L 10: Why don't you use simply "OC-M" as denominator? In addition, it is assumed here that OC*
*in free microaggregates and mineral fractions is older than OC in macroaggregates. Do you have*
*any data supporting this assumption? If not, you need to reformulate relevant sentences*
*throughout the manuscript.*

We agree with the editor that the total OC in M could have been used in the denominator (the sum
of each sub-fraction is equal to the total OC in macroaggregates). However, we feel that displaying
the different sub-fractions contained within the macroaggregates is a clearer way to present this
index.

The second question has been already answered in "major comments" point 1.

*P 7 L 17: 30 g soil "on a dry mass basis"?*

No, it was weighted on a fresh mass basis. It has been specified in the test (page 8, line 19).

*"P 7 L 14: There must have been significant reductions in soil moisture given the high incubation*
*temperature and 32 days of incubation. Please clarify this.*

As stated in the previous MS version (see lines 24-25 in page 8) the moisture content of the samples
was regularly checked for potential water losses by evaporation by weighting the bottles, but there
was not water losses and therefore it was not necessary to add any water". Please note that 32 days
of incubation is not a long period for water losses to occur.

*"P 8 L 1: Please describe why you opted for the nonparametric test. You might need to mention any*
*prior test for normal distribution".*

We used a non-parametric test because our sampling design was not balanced. That is, we did not
have the same amount of representative samples (nº of replicates) across eroding, transport and
depositional areas (see Table 1). Please note that prior normality distribution tests are not required
when non-parametric tests are performed.

*Sections 3.2-3.3: These two short sections can be better combined into one section, in a more*
*coherent way to compare OC fractions among the three watershed components.*

Ok, in agreement to the editor we have integrated 3.2 and 3.3

*Sections 3.4 & 3.5: Please also consider to integrate these sections in the preceding one or in a*
*separate section on OC quality.*

Ok, in agreement to the editor we have integrated 3.4 and 3.5 sections.

*"P 11 L 14: Please rephrase "macroaggregates are the nucleus for microaggregate formation".*
*How can larger macroaggregates function as the nucleus?"*

This sentence is based in Oades (1984) theory who postulated that "the roots and hyphae holding
together the macroaggregate form the nucleus for microaggregate formation in the center of the
macroaggregates. Other authors also explain this: "inside the macroaggregates, the presence of
decomposed organic matter, metabolites and biogenic products, polyvalent cations, and other
binding agents promoted the solid phase reaction between organic matter and clay and silt particles
leading to the formation of stable microaggregates (Edwards and Bremner, 1967; Golchin et al.,
1994). In addition in Six et al., 2004 there is a review of different models of soil aggregation where
this kind of hierarchical order of aggregation is included

*P 12 L 20-25: This type of 1 to 1 comparison between sediment and source soils does not make*
*sense, because three sources have different source capacity. Please take into consideration*
*estimates of source capacity in evaluating C enrichment or depletion during OC transport.*

In contrast with the editor view, we feel that comparing sediment with each one of the potential
sources, specially, at coarser scales, is a key aspect to determine the potential sources of eroded OC
in the sedimentary deposits through the catchment and also to progress in the knowledge on the
stabilization mechanisms in the redistribution of eroded carbon by water erosion processes.
Furthermore, our own experience in previous works (Boix Fayos et al., 2015, 2017) where a mean
value of OC was assumed as representative of the whole catchment sources can lead to an
overestimation of the role of mineralization during the redistribution of sediments when a very
plausible reason is that sediments come from low-OC sources as bank and channel sediments.

*P 13 L 4: Typo at the end of the sentence.*

It has been corrected

**- P 13 L 6: "sediments"? –**

We have rewritten the sentence (page 15, lines 1-2).

**P 13 L 12: "soil forming" or "aggregate forming"?**

We have changed soil forming by aggregate forming besides both terms have the same meaning
(page 15, line 9).

**"P 13 L 17-21: Please rewrite (better split) this long and vague sentence. This and the following**
**sentences are logically conflicting, because you are arguing that OC in the deposited sediments is**
**more stabilized than the source soils, even though more labile materials, as evidenced by higher**
**BR rates, exist in the same sediments. Please clarify this".**

Please note that our statements are not contradictory at all. On one hand, higher basal respiration is
a surrogate of microbial activity and therefore can be used as an index of the lability of the material
being degraded by the microorganisms. Microorganisms are activated by the presence of labile
(easily decomposable) organic matter and then respire more. But this labile organic matter is further
transformed in other bioproducts which can be more or less recalcitrant, can be chemically attached
to other molecules and even physically protected by aggregates, as it has been pointed out by many
previous studies (Six et al., 2000; 2002; Cotrufo et al 2013; 2015, Denef et al 2014, etc). Atthe end organic matter can be protected (and preserved) by their inherent chemical composition, by physical
protection mechanisms (intra-POM), or both. See Schmidt et al 2011 for futher details.

***P 13 L 21: typo "Thw"***

It has been changed (page 15, line 19).

**P 13 L 22: Please specify what you meant by "microbial induced processes"**

We have changed it to clarify (page 15, lines 21-22).

*"P 14 L 10: You did not measure "microbial activity".*

We have measured "Basal respiration" that is an estimate of the total microbial activity in soils
(Vanhala et al., 2005). Any way, we have change microbial activity by basal respiration rates to clarify
(page 16, line 8).

***P 14 L 16-18: Again, this short paragraph can find a better place in the preceding one***

Ok, it has been done (page 15, lines 15-18)

***P 15 L5: Again, you need to clarify how more stabilized OC in deposited sediments exhibited higher***
***rates of BR compared with those measured for the source soils. –***

It has been already clarified

**Fig 3: Please clarify in the figure legend whether significant differences indicated by different**
**letters are among the compared fractions or soil/sediment samples.**

Thanks for the suggestion. We agree that the Figure legend was not clear enough. Thus we have
changed it specifying  that the differences are among soils/sediment for each aggregate class. We
have changed it in Figure 3A, 3B, 4A and 4B (page 17, lines 16-23).

Elliott et al., 1991 has been now included in the reference list and Page 4, line 5 (suggesting by the
review 1)

We have also modified the contribution of each author (see Page 17, lines 16-23)

**References  mentioned in this response:**

**Almagro, M., López, J., Querejeta, J.I., Martínez-Mena, M. 2009. Temperature dependence of soil**
**CO2 efflux is strongly modulated by seasonal patterns of moisture availability in a Mediterranean**
**ecosystem. Soil Biology and Biochemistry 41, 594-605.**

**Berhe, A.A., Kleber, M.: Erosion, deposition, and the persistence of soil organic matter:**
**Mechanistic considerations and problems with terminology, Earth Surface Processes and**
**Landforms, 38, 908-912, doi: 10.1002/esp.3408, 2013.**

**Berhe, A.A., Barnes, R.T., Six, J., Marín-Spiotta, E.: Role of Soil Erosion in Biogeochemical Cycling of**
**Essential Elements: Carbon, Nitrogen, and Phosphorus,      Annual     Review     of     Earth     and**
**Planetary Sciences, 46, 521-548, doi: 10.1146/annurev-earth-082517-010018, 2018.**

Boix-Fayos, C., de Vente, J., Albaladejo, J., Martıíez-Mena, M., Soil carbon erosion and stock as
affected by land use changes at the catchment scale in Mediterranean ecosystems, Agriculture,
Ecosystems and Environment 133, 75–85, 2009

Boix-Fayos, C., Nadeu, E., Quiñonero, J.M., Martínez-Mena, M., Almagro, M., de Vente, J.:
Sediment flow paths and associated organic carbon dynamics across a Mediterranean catchment,
Hydrological Earth Systems Science, 19, 1209-1223, doi:10.5194/hess-19-1209, 2015.

Boix-Fayos, C., Martínez-Mena, M., Pérez Cutillas, P., de Vente, J., G. Barberá, G., Mosch, W.,
Navarro Cano, J. A., Gaspar, L., Navas, A. 2017. Carbon redistribution by erosion processes in an
intensively disturbed catchment. Catena, 149, 799-809

Conen, F., Zimmermann, M., Leifeld, J., Seth, B., Alewell, C., 2008. Relative stability of soil carbon
revealed by shifts in? 15 N and C: N ratio. Biogeosciences 5, 123e128.

Cotrufo, M.F., Wallenstein, M.D., Boot, C.M., Denef, K., Paul, E., 2013. The microbial efficiency-
matrix stabilization (MEMS) framework integrates plant litter decomposition with soil organic
matter stabilization: do labile plant inputs form stable soil organic matter? Global Change Biology
19, 988e995.

De Clercq, T., Heiling, M., Dercon, G., Resch, C., Aigner, M., Mayer, L. Mao, Y., Elsen, A., Steier, P.,
Leifeld, J., Merck, R.: Predicting soil organic matter stability in agricultural fields through carbon
and nitrogen stable isotopes. Soil Biology and Biochemistry, 88, 29-38, doi:
10.1016/j.soilbio.2015.05.011, 2015.

Denef, K.J., Six, J., Merckx, R., Paustian, K.: Carbon sequestration in microaggregates of no-tillage
soils with different clay mineralogy, Soil Science Society of America Journal, 68, 1935-1944, ISSN:
0361-5995, 1435-0661, 2004.

Edwards, A.P., Bremner, J.M., 1967. Microaggregates in soils. Journal of Soil Science 18, 64e73.

Garcia-Franco, N., Martínez-Mena, M., Goberna, M., Albaladejo, J.: Changes in soil aggregation and
microbial community structure control carbon sequestration after afforestation of semiarid
shrublands, Soil Biology & Biogeochemistry, 87, 110-121, doi:10.1016/j.soilbio.2015.04.012, 2015.

Golchin, A., Oades, J.M., Skjemstad, J.O., Clarke, P., 1994. Study of free and occluded particulate
organic matter in soils by solid state 13C P/MAS NMR spectroscopy and scanning electron
microscopy. Aust. J. Soil Res. 32, 285–309.

Hoffmann, T., Mudd, S. M., Van Oost, K., Verstraeten, G., Erkens, G., Lang, A., Middelkoop, H.,
Boyle, J., Kaplan, J. O., Willenbring, J., and Aalto, R.: Short Communication: Humans and the
missing C-sink: erosion and burial of soil carbon through time, Earth Surface Dynamics Discussions,
1, 93–112, doi:10.5194/esurf-1-45-2013, 2013.

JASTROW, J. D.: SOIL AGGREGATE FORMATION AND THE ACCRUAL OF PARTICULATE AND
MINERAL-ASSOCIATED ORGANIC MATTER, Soil Biol. Biochem., 28, 665-676, 1996

Lal, R.: Soil erosion and carbon dynamics. Soil and Tillage Research, 81, 137-142, doi:
10.1016/j.still.2004.09.002, 2005

**Nie, X., Li, Z., Huang, J., Liu, L., Xiao, H., Liu, C., Zeng, G.: Thermal stability of organic carbon in soil**
**aggregates as affected by soil erosion and deposition, Soil & Tillage Research, 175, 82–90, doi:**
**10.1016/j.still.2017.08.010, 2018.**

**Oades, J.M., 1984. Soil organic matter and structural stability: mechanisms and implications for**
**management. Plant and Soil 76, 319e337.**

**Pérez-Cutillas, P., Cataldo, M.F., Zema, D., de Vente, J., Boix-Fayos, C.: Efectos de la revegetación a**
**escala de cuenca sobre el caudal y la evapotranspiración en ambiente mediterráneo. Cuenca del**
**Taibilla (SE de España). Bosque 39, 119-129, doi: 10.4067/S0717-92002018000100011, 2018.**

**Quiñonero-Rubio, J.M., Nadeu, E., Boix-Fayos, C., de Vente, J: Evaluation of the Effectiveness of**
**Forest Restoration and Check-Dams to Reduce Catchment Sediment Yield, Land Degradation and**
**Development, 27, 1018-1031, doi:10.1002/ldr.2331, 2016.**

**Segoli, M., De Gryze, S., Dou, F., Lee, J., Post, W.M., Denef, K., Six, J., 2013. AggModel:a soil organic**
**matter model with measurable pools for use in incubation studies. Ecological Modelling 263, 1e9.**

**Six, J., Elliott, E.T., Paustian, K.: Soil structure and organic matter: I. Distribution of aggregate-size**
**classes and aggregate-associated carbon, Soil Science Society of American Journal, 64, 681–689,**
**doi: 10.2136/sssaj2000.642681x, 2000.**

**Six, R. T. Conant, E. A. Paul & K. Paustian. Stabilization mechanisms of soil organic matter:**
**Implications for C-saturation of soils J. Plant and Soil241:155–176, 2002.**

**Six, J., Bossuyt, H., Degryze, S., Denef, K.A.: history of research on the link between**
**(micro)aggregates, soil biota, and soil organic matter dynamics, Soil Tillage Research, 79, 7–31,**
**doi:10.1016/j.still.2004.03.008, 2004.**

**Schmidt MWI, Torn MS, Abiven S et al. (2011) Persistence of soil organic matter as an ecosystem**
**property. Nature, 478, 49–56.**

**Vanhala, P., Tamminen, P., Fritze, H.: Relationship between basal soil respiration rate, tree stand**
**and soil characteristics in boreal forests Environmental Monitoring and Assessment 101, 85-92,**
**2005.**

---

## Author Response (AR2)

CSIC
CONSEJO SUPERIOR DE INVESTIGACIONES CIENTÍFICAS

Murcia, Spain, 20th February 2019

Dear Dr. Ji-Hyung Park,

First of all, we would like to thank once again the Handling Editor for the time dedicated to our manuscript and for their numerous and very constructive and helpful comments. We have addressed all the comments from the Handling Editor and we feel that the manuscript has substantially improved with the changes introduced. We have included a point-by-point response to the comments by the Handling Editor, and here we summarize the mayor changes in this revised version of this manuscript.

According to the Handling Editor comment regarding our previous responses:

*"1. Inadequate author response*

"…Nevertheless, higher basal respiration in sediments compared to the source soils only occurred at the surface layer of the deposits (e.g., alluvial wedges and reservoir; please see Table 4), where at the same time aggregate formation is occurring (as explained before). On the contrary, at the deep layers of the alluvial wedges and the reservoir, where the carbon is being stored and stabilized, much lower basal respiration rates than at the upper layers, and even than in the source soils are observed.""

AND

*"Another related review comment and response:*

P 13 L 17-21: Please rewrite (better split) this long and vague sentence. This and the following sentences are logically conflicting, because you are arguing that OC in the deposited sediments is more stabilized than the source soils, even though more labile materials, as evidenced by higher BR rates, exist in the same sediments. Please clarify this.

"Please note that our statements are not contradictory at all. On one hand, higher basal respiration is a surrogate of microbial activity and therefore can be used as an index of the lability of the material being degraded by the microorganisms (Paul et al., 2001). Microorganisms are activated by the presence of labile (easily decomposable) organic matter and then respire more. But this labile organic matter is further transformed in other bioproducts which can be more or less recalcitrant, can be chemically attached to other molecules and even physically protected by aggregates, as it has been pointed out by many previous studies (Six et al., 2000; 2002; Cotrufo et al 2013; 2015, Denef et al 2014, etc). At the end organic matter can be protected (and preserved) by their inherent chemical composition, by physical protection mechanisms (intra-POM), or both. See Schmidt et al 2011 for futher details."

***Editor comment****: "As you noted, the surface and deep layers of both alluvial wedges and reservoir sediments exhibited different degrees of "microbial degradation" and "protection", which should be described and interpreted with caution. Please change ambiguous descriptions(e.g., L 33-35 on P 1: "Aggregate formation and OC accumulation, mainly*

*associated with macroaggregates and occluded microaggregates within macroaggregates,*
*were predominant in depositional areas".) throughout the manuscript and make it clear*
*that your argument would be valid for the deeper layers in depositional areas. Of course*
*you can cite some supporting papers in your discussion, but above all you need to base your*
*argument on your own findings. For instance, protected OC and microbial degradation*
*index values for surficial sediments (Table 4) don't support what you are saying here"*

**Response:** Following the Handling Editor suggestion, we have now clarified throughout the
manuscript (hereafter, MS) whether we are referring to superficial or deep layers of the
depositional sites. Moreover, the different role of both deposits in the OC mobilization by
erosion processes throughout the MS has been highlighted.

According with our results (explained in page 14, lines 20-24 in the previous MS version)
aggregate formation is occurring mainly at the surface layer of the depositional areas favored
by the growth of roots and the stimulation of microorganisms by root exudates of established
terrestrial and aquatic vegetation at these depositional areas.  In addition, in the upper
surface of the depositional areas an enrichment of the OC with respect to the most eroding
areas is also observed, which means than in these surface layers OC is being accumulated. At
the deep layers of the depositional areas, however, aggregate formation process unfolded very
slowly compared to the upper layers, and a more passive (resistant to oxidation) OC, more
preserved from mineralization (is also observed. This statement is also supported by the
observed  lower basal respiration rates in the deep layers than in the upper layers of the
depositional areas.

We thus consider that the two indices proposed in the previous version of the MS support our
interpretation. For example, the microbial degradation index is higher (OC less susceptible of
being degraded by microorganisms) at the deep than at the surface layers, which is consistent
with the higher content of the OC pool resistant to oxidation in the former than in the latter.
Likewise, the fact that the microbial degradation index was strongly negatively correlated
with basal respiration rates across the catchment support the reliability and validity of this
index. On the other hand, the OC stabilized (protected) in macroaggregates is higher at the
deep than at the surface layers, and even higher than that of the most eroded areas where the
sediment are coming. Moreover, at the surface layer of both depositional areas the proportion
of OC stabilized (protected) in macroaggregates is in the same range than that of the
sediment source areas, but higher than that obtained in the solids in suspension, in which
mineralization ratios are also higher and no aggregate formation is occurring. The fact that
an accumulation of OC is occurring in the surface layer of the depositional areas whereas the
degradation index is relatively low (*i.e.*, the OC is more susceptible to be degraded) is not
contradictory at all because our results suggest that the labile OC inputs (more easily
degradable, such as root exudates) are probably inductors of the formation of
microaggregates within macroaggregates. This statement is supported by the observed
positive correlations between the active OC pool (Mpom) and the percentages of
macroaggregates (M) and occluded microaggregates (Mm) and its OC contents (total OC-M
and OC-Mm) in these areas.

Nevertheless, and in agreement with the Handling Editor, we acknowledge that the microbial
degradation index might be used with caution since it would need to be combined with other
analytical approaches (such as stable isotopes analysis), which unfortunately have not been
conducted in this study, to enhance and reinforce its interpretation. Thus, and given that the
suppression of this index does not modify the interpretation of the results nor the main
conclusions drawn from this study, we have deleted it from the previous version of MS to
avoid misunderstandings and conflicting interpretations by the readers.

"2. Inadequate author response

"…We agree with the comment made by the editor that soil and sediment respiration rates
under field conditions may differ from those under laboratory controlled conditions."

***Editor comment:*** *Please provide this discussion in the main text and indicate clearly the*
*new discussion if you had already done it."*

**Response:**

Apologies beforehand, but we don't really understand the reason why the editor suggests to
include this issue in the discussion section since it was a methodological issue that had been
already addressed in the previous version of the manuscript. We acknowledge that we
misunderstood the Handling Editor's comment and emphasize the reason why we used
laboratory incubations rather that *in situ* field measurements to estimate organic matter
mineralization rates when a different concern (*i.e.*, if soil and sediment aerobic incubations
would be representative of anaerobic conditions of the deep layers at the depositional sites)
had been raised. Regarding this issue, and in agreement with the Handling Editor comment,
aerobic conditions can accelerate the degradation of organic matter in sediments otherwise
limited by $O_2$. But as previously explained, we needed to run laboratory incubations under
standardized conditions (e.g., temperature, moisture, $O_2$ availability) in order to make
mineralization rates comparable among sediment sources and deposits as well as to relate
them to OC quality. Please note that although our approach (*i.e.*, incubations under aerobic
conditions) would have slightly overestimated the mineralization rates of the sediments in
the deep layers at the alluvial wedges and the reservoir, such mineralization rates were still
about four-times lower than those from the surface sediments, which doesn't invalidate our
rationale. Nevertheless, and in order to better justify the proposed methodology, we have
now included a paragraph and provide references to justify our approach in section 2.4.

"3. Inadequate response to a specific comment:

P 2, L 1-2: Please be more specific in providing your major conclusion about the relative
importance of "temporary and permanent deposits. Do you mean that both sources are
equally important?

"According to the results from our work we consider that both, temporary and permanent
deposits are very important from the point of view of the physico-chemical mechanisms of
OC protection and stabilization that are occurring across the catchment. In addition, it is not
our purpose to determine if one of them is more important than the other. We consider that
both temporary and permanent deposits should be preserved, although not beyond the
natural fluvial dynamics specially for the natural transitory deposits (short-term residence
times), due to their high potential as C sinks."

***Editor comment:*** *Again, please pay attention to differences in basal respiration and*
*microbial degradation index among different sediment categories - i.e., the suspended/bed*
*sediments, and the surface and deep layers of depositional sediments. At least you need to*
*specify what role is important and how different it is among the sediment categories."*

**Response:** In agreement with the Handling Editor comment, we have removed these
general sentences from the abstract and the conclusion sections. Moreover, and as already
stated, we have now clarified when we are referring to superficial or deep layers of the
sediment deposits throughout the MS.

"4. Inadequate author response

P 7 L 10: Why don't you use simply "OC-M" as denominator? In addition, it is assumed here
that OC in free microaggregates and mineral fractions is older than OC in macroaggregates.
Do you have any data supporting this assumption? If not, you need to reformulate relevant
sentences throughout the manuscript.

"We agree with the editor that the total OC in M could have been used in the denominator
(the sum of each sub-fraction is equal to the total OC in macroaggregates). However, we feel
that displaying the different sub-fractions contained within the macroaggregates is a clearer
way to present this index."

*Editor comment: Please clarify why you felt so, or the index makes sense. Based on*
*what?"*

**Response:** As previously explained in the former response to the Handling Editor
comments, this index is based on a recent study by De Clercq et al., (2015). Nevertheless, as
already stated, we have dropped the microbial degradation index and all the interpretations
regarding this index from the revised version of the MS, as it has been explained in point 1.

"5. Inadequate author response:

P 7 L 14: There must have been significant reductions in soil moisture given the high
incubation temperature and 32 days of incubation. Please clarify this.

"As stated in the previous MS version (in page 8, lines 18-20) the moisture content of the
samples was regularly checked for potential water losses by evaporation by weighting the
bottles, but there was not water losses and therefore it was not necessary to add any water".
Please note that 32 days of incubation is not a long period for water losses to occur."

*Editor comment: This would be very difficult to believe for any research who has ever*
*done a similar soil incubation experiment. Where would evaporated water go during the*
*'long' period of 32 days? Did you keep the bottles closed or open to the air?*

As mentioned in the previous version of the MS (page 7, line 20) soil and sediment samples
were put "in hermetically-sealed flasks", so flasks were closed during the incubation
experiment, except after each measurement when flasks were opened for 30 minutes to avoid
the accumulation of $CO_2$. We have now clarified that flasks were opened for 30 minutes after
each measurement in the revised version of the manuscript since we had omitted this
important information. Apologizes for the missing information. Given the short time that
flasks were opened over the whole incubation period no water losses by evaporation occurred
(of course this was checked by monitoring gravimetric soil/sediment moisture periodically
after each measurement by weighting the flasks and comparing the weights after each
measurement with the initial weights).

*6. Minor correction needs:*

*-" Please check carefully editorial errors such as "100 cc volume" (P 5, L 13)"*

We have changed "cc" to "$cm^3$".

*-" P3 L 5 - P4 L2: Please check and split paragraph if required".*

The paragraph has been split as requested.

- *"P 8 L 1: Please describe why you opted for the nonparametric test. You might need to*
*mention any prior test for normal distribution".*

"We used a non-parametric test because our sampling design was not balanced. That is, we
did not have the same amount of representative samples (nº of replicates) across eroding,
transport and depositional areas (see Table 1). Please note that prior normality distribution
tests are not required when non-parametric tests are performed."

*Editor comment: Please describe this in section 2.5.*

Following the editor suggestion, we have now clarified why we opted for the nonparametric
test in the statistical analysis section from the revised version of the MS (page 8, lines 4-6)

- "P 11 L 14: Please rephrase "macroaggregates are the nucleus for microaggregate
formation". How can larger macroaggregates function as the nucleus?"

This sentence is based on Oades (1984) theory who postulated that "the roots and hyphae
holding together the macroaggregate form the nucleus for microaggregate formation in the
center of the macroaggregates".

*Editor comment: Please provide the reference, whenever you cite a specific paper.*

Please note that the reference by Oades (1984) had been already included in the main text as
well as in the reference list from the previous MS (page 12, line 12).

- Table 4: *"Protected OC" is still in use in the table though the caption uses the new term.*

We have changed "protected OC" to "stabilized OC in macroaggregates" in table 4 and
through the captions.

- *Figs. 3-4: Please double check different letters indicating significant differences. For*
*example, I doubt whether free mineral is really not different between bedload and surface*
*reservoir.*

Thank you very much for your deep review and apologizes for the mistake. We are so sorry.
You are right. The free mineral fraction is significantly different between the bedload and the
surface reservoir. We have now corrected ourselves and the letters indicating significant
differences among deposits have been changed in Figures 3 and 4.

Also, we have detected other errors in Figure 4A: the letters for the occluded mineral fraction
in suspended load and occluded microaggregates in the surface reservoir were missing; and
Figure 4B: letters for the occluded mineral fraction in forest soil was missing. These letters
have been now included.

A new reference from Paul E.A. 2001 (page 7, line 15) and Garcia-Franco 2015 has been now
included (now, 2015a and 2015b: page 11, lines 14 and 16).

Yours sincerely,

Dr. María Martínez-Mena, on behalf of all the co-authors

[revised manuscript text omitted]

Figure 1

Figure 2

[Figure]

Figure 3A

[Figure]

Figure 3B

[Figure]

Figure 4A

[Figure]

Figure 4B

[Figure]